# Predicting drug sensitivity of cancer cells based on DNA methylation levels

**Sofia P. Miranda**[1], **Fernanda A. Baião**[1], **Julia L. Fleck**[2]*, **Stephen R. Piccolo**[3]*

**1** Department of Industrial Engineering, Pontifical Catholic University of Rio de Janeiro, Rio de Janeiro, Rio de Janeiro, Brazil, **2** Mines Saint-Etienne, Univ Clermont Auvergne, CNRS, UMR 6158 LIMOS, Centre CIS, Saint-Etienne, France, **3** Department of Biology, Brigham Young University, Provo, Utah, United States of America

* julia.fleck@emse.fr (JLF); stephen_piccolo@byu.edu (SRP)

## Abstract

Cancer cell lines, which are cell cultures derived from tumor samples, represent one of the least expensive and most studied preclinical models for drug development. Accurately predicting drug responses for a given cell line based on molecular features may help to optimize drug-development pipelines and explain mechanisms behind treatment responses. In this study, we focus on DNA methylation profiles as one type of molecular feature that is known to drive tumorigenesis and modulate treatment responses. Using genome-wide, DNA methylation profiles from 987 cell lines in the Genomics of Drug Sensitivity in Cancer database, we used machine-learning algorithms to evaluate the potential to predict cytotoxic responses for eight anti-cancer drugs. We compared the performance of five classification algorithms and four regression algorithms representing diverse methodologies, including tree-, probability-, kernel-, ensemble-, and distance-based approaches. We artificially subsampled the data to varying degrees, aiming to understand whether training based on relatively extreme outcomes would yield improved performance. When using classification or regression algorithms to predict discrete or continuous responses, respectively, we consistently observed excellent predictive performance when the training and test sets consisted of cell-line data. Classification algorithms performed best when we trained the models using cell lines with relatively extreme drug-response values, attaining area-under-the-receiver-operating-characteristic-curve values as high as 0.97. The regression algorithms performed best when we trained the models using the full range of drug-response values, although this depended on the performance metrics we used. Finally, we used patient data from The Cancer Genome Atlas to evaluate the feasibility of classifying clinical responses for human tumors based on models derived from cell lines. Generally, the algorithms were unable to identify patterns that predicted patient responses reliably; however, predictions by the Random Forests algorithm were significantly correlated with Temozolomide responses for low-grade gliomas.

**Data Availability Statement:** Scripts for downloading and processing the data can be found at https://osf.io/r7pdb/.

**Funding:** This work was supported in part by the Coordination for the Improvement of Higher

Education Personnel (CAPES) - Finance Code 001.
The funders had no role in study design, data
collection and analysis, decision to publish, or
preparation of the manuscript.

**Competing interests:** The authors have declared
that no competing interests exist.

**Abbreviations:** ACC, Accuracy; AUC, Area under
the receiver operating characteristic curve; FDR,
Benjamini-Hochberg False Discovery Rate; GDSC,
Genomics of Drug Sensitivity in Cancer; KNN, K-
nearest neighbors; MAE, Mean absolute error;
MCC, Matthews correlation coefficient; MMCE,
Mean misclassification error; MSE, Mean squared
error; NB, Naïve Bayes; RF, Random forests;
RMSE, Root-mean-square deviation; SCC,
Spearman's rank correlation coefficient; SVM,
Support vector machines; TCGA, The Cancer
Genome Atlas; XGBoost, Extreme Gradient
Boosting.

## Introduction

Cancers are complex, dynamic diseases characterized by aberrant cellular processes such as excessive proliferation, resistance to apoptosis, and genomic instability [1]. Tumors are caused by somatic variations, which can affect individual nucleotides or larger segments of DNA [2]. Dysregulation of cellular function can also be caused by epigenetic modifications, including aberrant DNA methylation [3]. One goal of cancer research is to advance precision medicine through identifying genomic and epigenomic features that influence treatment outcomes in individuals [4]. In this context, therapeutic decisions have the potential to be guided by molecular signatures.

Cancer cell lines are cell cultures derived from tumor samples. They represent one of the least expensive and most studied preclinical models [5]. Drug screening in cell lines can be used to prioritize candidate drugs for testing in humans. In performing a screen, researchers calculate $IC_{50}$ values, which quantify the amount of drug necessary to induce a biological response in half of the cells tested for a given experiment [6]. Drugs with a relatively high potency (corresponding to low log-transformed $IC_{50}$ values) are generally considered to be the strongest candidates for use in humans, although patient safety must also be evaluated. After a candidate drug has been identified, researchers may seek to identify molecular markers associated with those responses, comparing cell lines that respond to the drug against those that do not. Such markers might be useful for elucidating drug mechanisms or eventually predicting clinical responses in patients [7].

Over the past decade, researchers have catalogued the molecular profiles of more than a thousand cancer cell lines and their responses to hundreds of drugs [8–10]. These resources have been made publicly available, thus providing an opportunity for researchers to identify molecular signatures that predict drug responses in a preclinical setting. In addition, recent efforts to catalog molecular profiles in human tumors have resulted in massive collections of publicly available molecular data for tumor samples [11–13]. Such data can be used to validate findings from preclinical studies and assess our ability to classify cancer patients into groups that will most likely benefit from a certain treatment [14].

Many computational methods have been proposed to predict anticancer drug sensitivity based on genetic, genomic, or epigenomic features of cancer cell lines. The most common approach is to generate a drug-specific model, which is independently trained using molecular observations and drug-response data from cell lines tested with each drug individually. Linear-regression based, drug-specific models have been developed using gene expression data [7, 8, 15] or a combination of gene expression data and other genomic data types, such as copy number alterations and DNA methylation [16]. Non-linear models using a single data type or multiple data types have also been proposed, including artificial neural networks, random forests, support vector machines (SVM), kernel regression, latent and Bayesian approaches, attractor landscape analysis of network dynamics, unsupervised pathway activity models, and recommender systems [17–34]. Transfer-learning techniques have also been proposed to improve drug-response prediction performance for one type of cancer by incorporating data from other types of cancer [35]. Drug response information has also been modeled in combination with chemical drug properties using elastic net regression, support vector machines, regularized matrix factorization, and manifold Learning [36–40].

Most recent cell-line studies have emphasized the potential to predict drug responses based on gene-expression profiles [17, 41–44]. Technologies for profiling gene-expression levels are widely available and reflect the downstream effects of genomic and epigenomic aberrations. However, gene-expression profiles may be difficult to apply in the clinic because of the instability of RNA [45]. Moreover, gene-expression data are generated using a wide range of

technologies (e.g., different types of oligonucleotide microarrays and RNA-sequencing), and are preprocessed using diverse algorithms. Thus, it is often difficult to combine datasets from multiple sources (e.g., preclinical and tumor data). In this study, we focus on DNA methylation profiles, using cell-line data from the Genomics of Drug Sensitivity in Cancer (GDSC) database [7] in combination with tumor data from The Cancer Genome Atlas (TCGA) [46]. These projects used the same technology to quantify methylation levels, and the GDSC team created a version of the methylation data that had been normalized in a consistent manner, thus enabling us to perform a more systematic evaluation of whether DNA methylation levels can predict drug responses.

DNA methylation is an epigenetic mechanism that controls gene-expression levels. The addition of a methyl group to DNA may lead to changes in DNA stability, chromatin structure, and DNA-protein interactions. Hypermethylation of CpG islands in promoter regions of DNA has been acknowledged as an important means of gene inactivation, and its occurrence has been detected in almost all types of human tumors [47]. Similar to genetic alterations, methylation changes to DNA may alter a gene's behavior. However, hypermethylation can be reversed with the use of targeted therapy [48], making it an attractive target for anticancer therapy [49, 50].

In some cases, DNA methylation levels for a single gene may control cellular responses for a given drug. For example, MGMT hypermethylation predicts temozolomide responses in glioblastomas [51], and BRCA1 hypermethylation predicts responses to poly ADP ribose polymerase inhibitors in breast carcinomas [52]. However, in many cases, drug responses are likely influenced by the combined effects of many genes interacting in the context of signaling pathways [53]. Accordingly, to maximize our ability to predict drug responses, it is critical to account for this complexity.

In this study, we use DNA methylation profiles from preclinical samples to model drug responses for eight anti-cancer drugs. We compare the performance of five classification algorithms and four regression algorithms that encompass a diverse range of methodologies, including tree-based, probability-based, kernel-based, ensemble-based, and distance-based approaches. We use classical algorithms as a way to establish a performance baseline against which other algorithms might be compared when working with DNA methylation profiles. For regression, we predict $IC_{50}$ values directly. For classification, we use discretized $IC_{50}$ values. For both types of algorithm, we artificially subsample the data to varying degrees to evaluate whether training models based on relatively extreme outcomes would yield improved performance; we assess our ability to predict drug responses using as few as 10% of the cell lines (those with the most extreme $IC_{50}$ values). An underlying motivation of this approach was to decrease data-generation costs. For example, if it could be shown that generating data for relatively few (extreme) responders performs as well as or better than generating data for responders across the full range of response values, cost savings may result. Perhaps surprisingly, the classification algorithms performed best when only 10–20% of the cell lines were used. The regression algorithms performed best when we trained the models using the full range of drug-response values, although this depended on the performance metrics we used. Finally, we derived classification models from the cell-line data and predicted drug responses for TCGA patients. In most cases, the models failed to generalize effectively; however, predictions by the Random Forests algorithm were significantly correlated with Temozolomide responses for low-grade gliomas.

## Methods

The GDSC database contains data for human cell lines derived from common and rare types of adult and childhood cancers. GDSC provides multiple types of molecular data for these cell

lines in addition to response values for 265 anti-cancer drugs. In this work, we used database version GDSC1, which includes data for 987 cell lines curated between 2010 and 2015 [7]. Drug responses were measured as the natural log of the fitted $IC_{50}$ value. The more sensitive the cell line, the lower the $IC_{50}$ value for any given drug. We developed machine-learning models of drug response using DNA methylation data from GDSC1 that had been preprocessed and summarized as gene-level *beta* values [7]; these values ranged between 0 and 1 (higher values indicated relatively high methylation for a given gene). We used all available methylation regions, represented by gene-level summarized values, as input to the classification and regression algorithms.

For external validation, we used DNA methylation data and clinical drug-response values from TCGA. We selected eight drugs that were administered to TCGA patients and present in GDSC: Gefitinib, Cisplatin, Docetaxel, Doxorubicin, Etoposide, Gemcitabine, Paclitaxel, and Temozolomide. These drugs represent a variety of molecular mechanisms, including DNA crosslinking, microtubule stabilization, and pyrimidine anti-metabolization. Aside from Gefitinib, which we used for model optimization on GDSC data, these drugs were associated with the largest number of patient drug-response values in TCGA [54]. GDSC provides DNA methylation values for 6,035 TCGA samples that had been preprocessed using the same pipeline as the GDSC samples. We obtained drug-response data for TCGA patients from [55].

Cell lines with missing $IC_{50}$ values were excluded on a per-drug basis; thus, sample sizes differed across the drugs. We applied Z-score normalization on a per-gene basis across all samples in GDSC and TCGA. Next, we used ComBat [56] to adjust for systematic differences between the two datasets (GDSC and TCGA); we also specified cell type as a covariate to adjust for methylation patterns associated with this factor.

We started with a classification analysis. Classification algorithms are widely available, and their predictions are intuitive to interpret—they assign probabilities to each sample for each class. To enable classification for the GDSC cell lines, we discretized the $IC_{50}$ values into "low" and "high" values. However, the choice of a threshold for distinguishing low and high values was necessarily arbitrary. Initially, we used the median $IC_{50}$ value across all cell lines as a threshold. However, cell lines with an $IC_{50}$ just above or below this threshold naturally showed very little difference in their drug responses, even though they were assigned to different classes. In contrast, cell lines with extreme $IC_{50}$ values (far from the threshold) had much more distinct drug responses. To investigate the effects of using a threshold to discretize the $IC_{50}$ values for classification, we used subsampling. We created 10 different scenarios that included increasing percentages of the overall data. First, we sorted the samples by $IC_{50}$ value in ascending order. For the first scenario, we evaluated cell lines with the 5% lowest and 5% highest $IC_{50}$ values (10% of the total data). In the next scenario, we evaluated cell lines with the 10% lowest and 10% highest $IC_{50}$ values (20% of the total data), and so on. The last scenario included all the data, where the lowest 50% were considered to have low $IC_{50}$ values and the highest 50% were considered to have high values (S1 Fig). For the regression analysis, we followed a similar process for subsampling but retained the continuous nature of the $IC_{50}$ values.

For both classification and regression, we used the Random Forests (tree-based) [57], (Support Vector Machines (kernel-based) [58], Gradient Boosting Machines (ensemble-based) [59], and k-Nearest Neighbors (distance-based) [60] algorithms. We used the Naïve Bayes (probability-based) [61] algorithm for classification but not for regression because this algorithm is only designed for classification analyses. We performed the analyses using the R programming language [62] and Rstudio (https://rstudio.com). The machine-learning algorithms were implemented in the following R packages: mlr [63], e1071 [64], xgboost [65], randomForest [66], and kknn [67].

Using the GDSC cell-line data, we sought to select the best hyperparameters for each algorithm via nested cross validation. We used the *mlr* package [63] to randomly assign the cell

**Table 1. Descriptions of the algorithms we tested and hyperparameters that we evaluated via nested cross validation.** Hyperparameter optimization was performed for all tested algorithms. All parameter combinations for each algorithm were evaluated via nested cross validation; optimal combinations were then used for outer-fold predictions.

| Algorithm | Hyperparameters | Definition | Tested Values |
|---|---|---|---|
| classif.svm and regr.svm | 1. Kernel | The kernel function used to transform data to higher-dimensional spaces and then become linearly separable. | Linear; Radial; Polynomial; Sigmoid |
| | 2. Cost | The regularization parameter in the cost function, to penalize missing classifications. | 0.1; 1; 10; 100 |
| | 3. Scale | Whether the variables should be scaled. | True; False |
| classif.randomForest and regr. randomForest | 1. Ntree | The number of trees to grow. | 100; 500; 1000 |
| | 2. Nodesize | Minimum size of terminal nodes. | 1; 3; 5; 7 |
| | 3. Importance | Whether the importance of predictors should be assessed. | True; False |
| classif.kknn and regr.kknn | 1. K | The number of neighbors considered. | 3; 7; 10 |
| | 2. Scale | Whether to scale variables to have equal standard deviation. | True; False |
| classif.naiveBayes | 1. Laplace | The amount of Laplace (additive) smoothing. | 0; 1; 5; 10 |
| classif.xgboost | 1. Nround | The maximum number of boosting iterations. | 100; 250; 500 |
| | 2. Max_depth | The maximum depth of a tree. | 1; 5; 10 |
| | 3. Eta | How much the contribution of each tree is scaled to the overall approximation, to control the learning rate. | 0.1; 0.3; 0.5 |
| regr.xgboost | 1. Nround | The maximum number of boosting iterations. | 100; 250; 500 |
| | 2. Eta | How much the contribution of each tree is scaled to the overall approximation, to control the learning rate. | 0.1; 0.3; 0.5 |

lines to 10 outer folds and 5 inner folds (per outer fold). For each combination of algorithm and data-subsampling scenario, we evaluated the performance of all hyperparameter combinations (Table 1) using the inner folds; we used MMCE (Mean Misclassification Error) [68] for classification and MSE (Mean Squared Error) [69] for regression as evaluation metrics in the inner folds (defaults in *mlr*). For the outer-fold predictions, we assessed performance for predicting drug responses using several performance metrics. This enabled us to evaluate how consistently the algorithms performed. For the classification analysis, we used accuracy (1—MMCE), area under the receiver operating characteristic curve (AUC) [70], F1 measure [71], Matthews correlation coefficient (MCC) [72], recall, and specificity. For the regression analysis, we used Mean Absolute Error (MAE), Root Mean Square Error (RMSE) [69], R-squared coefficient of determination [73] and Spearman's rank correlation coefficient (SCC) [74].

After assessing the algorithms separately for the classification and regression approaches, we evaluated the predictive ability of these two types of tasks against one another. We calculated the Spearman correlation coefficient as a nonparametric measure of the concordance between the predicted probabilities (classification algorithms) and predicted $IC_{50}$ values (regression algorithms).

For the classification and regression analyses, we used feature selection to identify genes deemed to be most informative. We performed an information-gain analysis, assigning an importance score to each feature (gene). More specifically, we estimated the relative importance of each gene based on the conditional entropy of the class variable with respect to that gene. Entropy measures the amount of randomness in the information. Thus, higher information gain implies lower entropy. This analysis was implemented using the FSelectorRcpp package [75]. To assess the functional relevance of the top-ranked genes, we used a gene-set overlap technique implemented in the Molecular Signatures Database 3.0 [76]. As candidate gene sets, we included the *C2 (curated gene sets)*, *C4 (computational genes sets)*, and *C6 (oncogenic signature gene sets)*. We used a False Discovery Rate q-value threshold of 0.05.

For additional validation, we trained classification models based on discretized drug responses in the GDSC cell lines and then predicted patient drug responses using tumor data from TCGA. These patient responses were based on clinical data, having no direct relation to $IC_{50}$ values. Because the patient-response values were categorical in nature, we only performed classification for these data. We used nested cross validation to perform hyperparameter optimization using the GDSC (training) data. To evaluate the relationship between the predicted labels and actual clinical responses, we calculated Spearman's rank correlation coefficient and a corresponding p-value for each combination of algorithm and data-subsampling scenario; then we used the Benjamini-Hochberg False Discovery Rate to adjust for multiple tests [77].

## Results

Using data from 987 cell lines, we used machine-learning algorithms to evaluate the potential to predict cytotoxic responses based on genome-wide, DNA methylation profiles. Second, we examined which genes were most predictive of these responses. Finally, we evaluated the feasibility of predicting clinical responses in humans based on models derived from cell-line data.

### Classification analysis using cell-line data

We collected DNA methylation data and $IC_{50}$ response values for eight drugs from the GDSC repository. In our initial analysis, we aimed to predict categories (classes) of drug sensitivity. These categories represented whether each cell line exhibited a "low" or "high" response to each drug, corresponding to relatively low or high $IC_{50}$ values, respectively. This categorization facilitated a simplified yet intuitive interpretation of the treatment outcomes and enabled us to use classification algorithms, which have been implemented for a broader range of algorithmic methodologies than regression algorithms.

Before performing classification, we categorized each cell line on a per-drug basis, according to whether its $IC_{50}$ value was greater than the median across all cell lines. One limitation of categorizing the cell lines in this way was that cell lines just above or below the median threshold showed a relatively small difference in $IC_{50}$ values, even though they were assigned to different classes. Generally, $IC_{50}$ values did not follow a multimodal distribution (Fig 1). Therefore, we evaluated whether classification performance could be improved by excluding cell lines with an $IC_{50}$ value relatively close to the median, even though this would reduce the amount of data available for training and testing. We evaluated ten scenarios that varied the number of cell lines used. In the most extreme scenario, we used methylation data for cell lines with the 5% lowest and 5% highest $IC_{50}$ values. In describing these subsampling scenarios, we use a notation that indicates the percentage of samples on each side of the distribution as well as the algorithm type. For example, when we analyzed the samples with the 5% highest and 5% lowest $IC_{50}$ values and employed a classification algorithm, we indicate this using "+-5%c". The equivalent scenario for regression was represented as +-5%r.

We evaluated the performance of five classification algorithms using six performance metrics (see Methods). In addition, we optimized hyperparameters via nested cross validation; Table 1 lists the hyperparameters we evaluated. Initially, we evaluated Gefitinib, an EGFR inhibitor. Overall, the algorithms performed best when relatively few cell lines (+-5%c and +-10%c) were used to train and test the models, attaining area-under-the-receiver-operating-characteristic curve (AUC) and classification-accuracy values as high as 0.93 and 0.84 (Table 2). This pattern was consistent across all five algorithms and all six metrics that we evaluated (Fig 2). However, the SVM algorithm consistently achieved higher classification performance than the other algorithms for this drug.

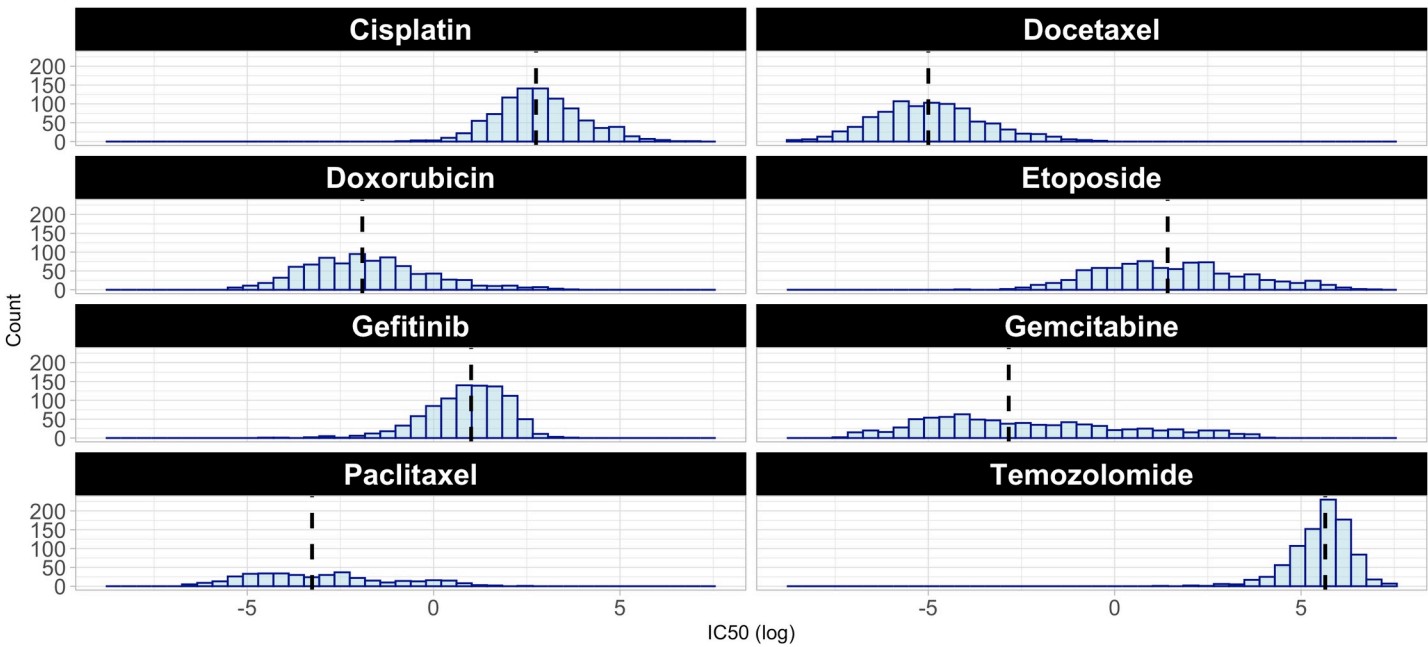

**Fig 1. Histograms for each drug based on drug response (IC$_{50}$ values) for the GDSC dataset.** The black line represents the median value for each subsample across all available cell lines for each drug.

When evaluating the seven remaining drugs, we continued to see a trend in which using a relatively small proportion of the data resulted in better classification performance. For Cisplatin, Docetaxel, Doxorubicin, and Etoposide, the best performance was attained for +-5%c and +-10%c, and the best-performing algorithms were always SVM or Random Forests (RF) (S1–S7 Tables). In contrast, for Gemcitabine, the highest AUC value (0.82) was obtained for +-15%c (SVM algorithm). For Paclitaxel, the Random Forests algorithm performed best for +-10%c (AUC = 0.75). The overall highest AUC value was attained for Docetaxel (0.97, +-10%c, Random Forests and SVM). S2–S8 Figs illustrate these results across all algorithms, metrics, and drugs and show that generally the top-performing algorithms were consistent across all metrics, although these patterns were less consistent in scenarios where the highest AUC values were lower than 0.80.

To further analyze combinations of subsampling scenarios and classification algorithms, we ranked the AUC values for all combinations and for each drug (where the lowest rank was considered best and represented the highest AUC value). Subsequently, we calculated the average AUC rank across all drugs. The best performance was attained for +-10%c (SVM) and +-10%c (Random Forests), achieving average ranks of 4.75 and 5.13, respectively (Table 3). When we evaluated the minimum, mean, and maximum AUC values for each combination of drug and algorithm, Docetaxel attained the best overall performance (Table 4).

### Regression analysis using cell-line data

We performed a regression analysis using the same DNA methylation data but with continuous IC$_{50}$ response values for the same eight drugs. For this analysis, we applied four regression algorithms and evaluated their performance using nested cross validation and four performance metrics (RMSE, MAE, R-squared and SCC). As with the classification analysis, we performed data subsampling to evaluate the effects of using relatively extreme IC$_{50}$ values. For Gefitinib and the MAE and RMSE metrics, all algorithms performed best when all cell lines

**Table 2. Classification results for all subsampling scenarios and algorithms for Gefitinib.**

| Scenario | Method | ACC | AUC | F1 | MCC | Recall | Specificity |
|---|---|---|---|---|---|---|---|
| +-5%c | SVM | 0.82 | **0.93** | 0.80 | 0.65 | 0.85 | 0.78 |
| +-5%c | Random Forest | 0.82 | 0.82 | **0.82** | 0.66 | **0.89** | 0.74 |
| +-5%c | KNN | 0.72 | 0.84 | 0.67 | 0.45 | 0.63 | 0.80 |
| +-5%c | XGBoost | 0.77 | 0.83 | 0.75 | 0.54 | 0.76 | 0.78 |
| +-5%c | Naive Bayes | 0.73 | 0.74 | 0.73 | 0.45 | 0.76 | 0.70 |
| +-10%c | SVM | **0.84** | 0.92 | **0.82** | 0.69 | 0.85 | **0.83** |
| +-10%c | Random Forest | 0.80 | 0.89 | 0.79 | 0.61 | 0.84 | 0.77 |
| +-10%c | KNN | 0.75 | 0.86 | 0.71 | 0.49 | 0.68 | **0.83** |
| +-10%c | XGBoost | 0.78 | 0.88 | 0.77 | 0.56 | 0.80 | 0.75 |
| +-10%c | Naive Bayes | 0.68 | 0.69 | 0.66 | 0.35 | 0.68 | 0.67 |
| +-15%c | SVM | 0.81 | 0.86 | 0.81 | 0.63 | 0.83 | 0.79 |
| +-15%c | Random Forest | 0.75 | 0.84 | 0.75 | 0.50 | 0.78 | 0.71 |
| +-15%c | KNN | 0.72 | 0.79 | 0.71 | 0.45 | 0.71 | 0.73 |
| +-15%c | XGBoost | 0.74 | 0.83 | 0.75 | 0.51 | 0.80 | 0.68 |
| +-15%c | Naive Bayes | 0.66 | 0.66 | 0.68 | 0.32 | 0.76 | 0.56 |
| +-20%c | SVM | 0.75 | 0.83 | 0.75 | 0.51 | 0.77 | 0.73 |
| +-20%c | Random Forest | 0.72 | 0.80 | 0.73 | 0.44 | 0.76 | 0.69 |
| +-20%c | KNN | 0.68 | 0.78 | 0.69 | 0.37 | 0.71 | 0.66 |
| +-20%c | XGBoost | 0.72 | 0.80 | 0.73 | 0.44 | 0.76 | 0.69 |
| +-20%c | Naive Bayes | 0.64 | 0.64 | 0.68 | 0.28 | 0.79 | 0.48 |
| +-25%c | SVM | 0.74 | 0.81 | 0.75 | 0.48 | 0.78 | 0.70 |
| +-25%c | Random Forest | 0.72 | 0.79 | 0.74 | 0.45 | 0.79 | 0.66 |
| +-25%c | KNN | 0.70 | 0.77 | 0.71 | 0.41 | 0.73 | 0.68 |
| +-25%c | XGBoost | 0.72 | 0.79 | 0.72 | 0.43 | 0.74 | 0.70 |
| +-25%c | Naive Bayes | 0.60 | 0.62 | 0.67 | 0.23 | 0.80 | 0.41 |
| +-30%c | SVM | 0.72 | 0.78 | 0.74 | 0.45 | 0.78 | 0.66 |
| +-30%c | Random Forest | 0.69 | 0.75 | 0.70 | 0.38 | 0.74 | 0.63 |
| +-30%c | KNN | 0.68 | 0.75 | 0.70 | 0.37 | 0.74 | 0.63 |
| +-30%c | XGBoost | 0.69 | 0.77 | 0.70 | 0.38 | 0.74 | 0.63 |
| +-30%c | Naive Bayes | 0.60 | 0.60 | 0.66 | 0.21 | 0.79 | 0.41 |
| +-35%c | SVM | 0.68 | 0.76 | 0.70 | 0.37 | 0.72 | 0.64 |
| +-35%c | Random Forest | 0.67 | 0.73 | 0.69 | 0.34 | 0.73 | 0.60 |
| +-35%c | KNN | 0.67 | 0.71 | 0.68 | 0.34 | 0.70 | 0.64 |
| +-35%c | XGBoost | 0.66 | 0.70 | 0.67 | 0.32 | 0.69 | 0.62 |
| +-35%c | Naive Bayes | 0.59 | 0.60 | 0.66 | 0.20 | 0.79 | 0.40 |
| +-40%c | SVM | 0.67 | 0.73 | 0.68 | 0.35 | 0.71 | 0.63 |
| +-40%c | Random Forest | 0.65 | 0.71 | 0.67 | 0.30 | 0.71 | 0.58 |
| +-40%c | KNN | 0.60 | 0.66 | 0.61 | 0.21 | 0.64 | 0.57 |
| +-40%c | XGBoost | 0.65 | 0.70 | 0.65 | 0.29 | 0.68 | 0.61 |
| +-40%c | Naive Bayes | 0.57 | 0.58 | 0.64 | 0.16 | 0.78 | 0.36 |
| +-45%c | SVM | 0.67 | 0.72 | 0.69 | 0.35 | 0.72 | 0.62 |
| +-45%c | Random Forest | 0.64 | 0.70 | 0.66 | 0.30 | 0.71 | 0.57 |
| +-45%c | KNN | 0.63 | 0.66 | 0.64 | 0.26 | 0.66 | 0.60 |
| +-45%c | XGBoost | 0.65 | 0.69 | 0.65 | 0.31 | 0.67 | 0.62 |
| +-45%c | Naive Bayes | 0.58 | 0.59 | 0.65 | 0.18 | 0.78 | 0.39 |
| +-50%c | SVM | 0.65 | 0.70 | 0.66 | 0.30 | 0.70 | 0.60 |
| +-50%c | Random Forest | 0.64 | 0.69 | 0.66 | 0.29 | 0.70 | 0.59 |

*(Continued)*

**Table 2.** (Continued)

| Scenario | Method | ACC | AUC | F1 | MCC | Recall | Specificity |
|----------|--------|-----|-----|-----|-----|--------|-------------|
| +-50%c | KNN | 0.60 | 0.65 | 0.60 | 0.20 | 0.61 | 0.59 |
| +-50%c | XGBoost | 0.63 | 0.68 | 0.64 | 0.27 | 0.65 | 0.62 |
| +-50%c | Naive Bayes | 0.58 | 0.59 | 0.64 | 0.17 | 0.77 | 0.39 |

Bold font indicates the best-performing combination for each metric.

were used to train and test the models, attaining RMSE values as low as 0.95 (lower is better, see Table 5). However, for the R-squared and SCC metrics, the +-5%r subsampling scenario resulted in the best performance in some cases. Typically, the magnitude of the differences between the original and predicted $IC_{50}$ values was larger toward the extremes, resulting in relatively high MAE and RMSE values when middle values were excluded. In contrast, SCC is a rank-based metric, and the algorithms struggled most to differentiate between $IC_{50}$ values toward the middle of the distribution. We observed similar patterns for the other seven drugs (S8–S14 Tables).

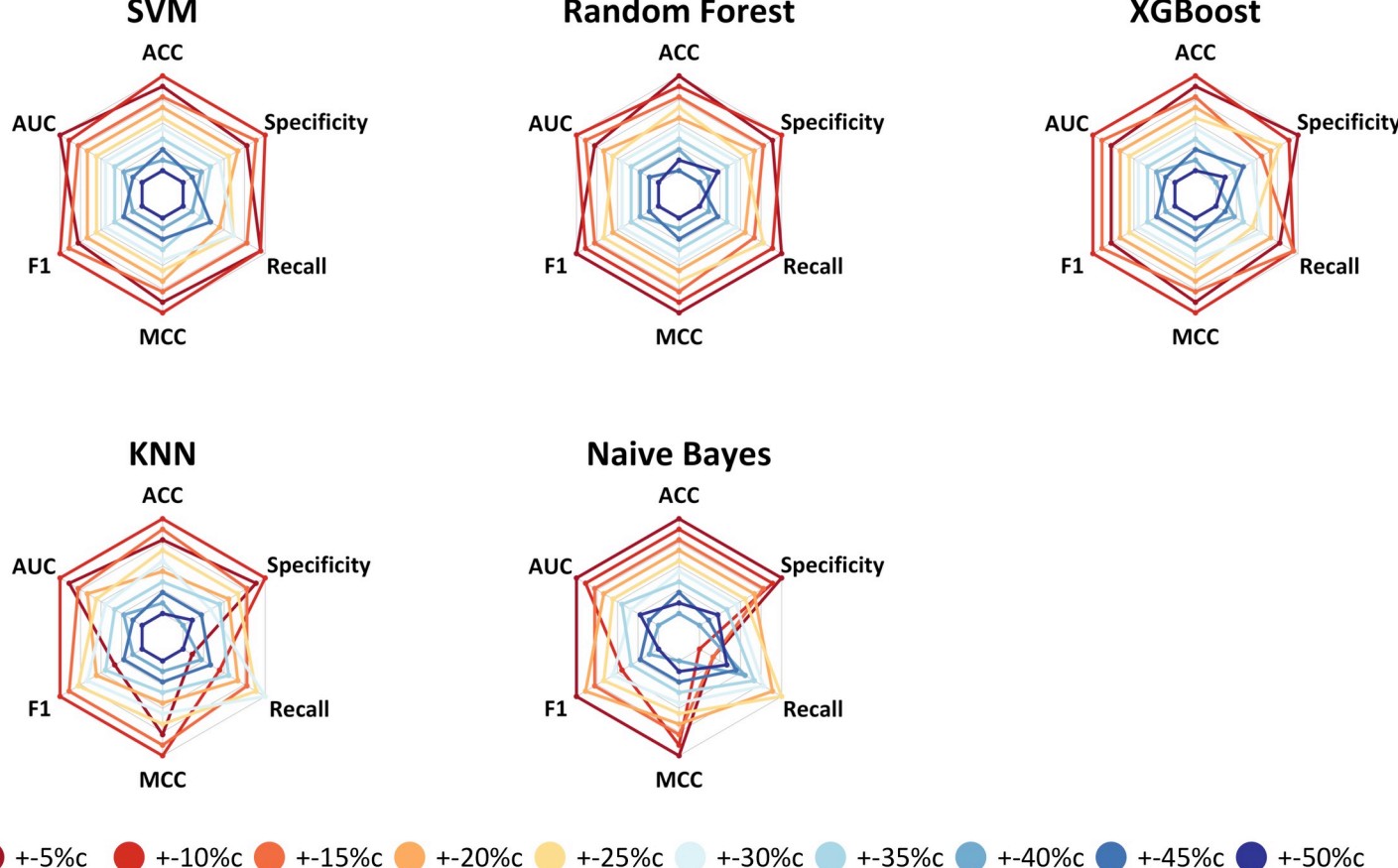

**Fig 2. Gefitinib classification results across six metrics.** These "spider" graphs illustrate how each classification algorithm performed in each subsampling scenario via cross validation on the GDSC cell-line data. Results that are further away from the center represent higher metric values (relatively better performance) than results closer to it. These metrics are accuracy (ACC), specificity, recall, Matthews correlation coefficient (MCC), F1 score (F1) and area under the receiver operating characteristic curve (AUC). Scenarios that used relatively few cell lines—but those with the most extreme $IC_{50}$ values—performed best for all algorithms. Specific metric values may be found in Table 2.

**Table 3. Summary of AUC values across all combinations of subsampling scenario and algorithm.** We ranked the AUC values for each combination and then calculated the average rank across the combinations (lower ranks imply better performance). In addition, this table lists the minimum, maximum, and standard deviation AUC value across the combinations.

| Scenario | Method | Average AUC Rank | Min AUC Value | Max AUC Value | Standard Deviation AUC Value |
|---|---|---|---|---|---|
| +-10%c | Random Forest | 4.75 | 0.72 | 0.97 | 0.08 |
| +-10%c | SVM | 5.13 | 0.65 | 0.97 | 0.10 |
| +-5%c | SVM | 5.14 | 0.74 | 0.95 | 0.08 |
| +-15%c | XGBoost | 7.50 | 0.68 | 0.94 | 0.09 |
| +-15%c | SVM | 7.63 | 0.66 | 0.93 | 0.10 |
| +-5%c | Random Forest | 7.71 | 0.77 | 0.93 | 0.06 |
| +-5%c | XGBoost | 7.86 | 0.69 | 0.96 | 0.09 |
| +-15%c | Random Forest | 9.13 | 0.70 | 0.92 | 0.08 |
| +-10%c | XGBoost | 10.13 | 0.58 | 0.94 | 0.12 |
| +-20%c | SVM | 10.75 | 0.66 | 0.92 | 0.09 |
| +-25%c | SVM | 11.25 | 0.70 | 0.90 | 0.07 |
| +-10%c | KNN | 12.75 | 0.67 | 0.91 | 0.09 |
| +-5%c | KNN | 13.14 | 0.69 | 0.92 | 0.07 |
| +-25%c | XGBoost | 15.38 | 0.67 | 0.89 | 0.07 |
| +-20%c | XGBoost | 15.88 | 0.65 | 0.91 | 0.09 |
| +-20%c | Random Forest | 16.00 | 0.64 | 0.91 | 0.08 |
| +-30%c | SVM | 16.25 | 0.68 | 0.86 | 0.06 |
| +-25%c | Random Forest | 16.50 | 0.70 | 0.88 | 0.07 |
| +-35%c | SVM | 19.00 | 0.68 | 0.84 | 0.05 |
| +-30%c | XGBoost | 20.50 | 0.62 | 0.87 | 0.08 |
| +-30%c | Random Forest | 20.63 | 0.65 | 0.85 | 0.07 |
| +-15%c | KNN | 21.25 | 0.61 | 0.87 | 0.10 |
| +-20%c | KNN | 23.38 | 0.63 | 0.88 | 0.09 |
| +-35%c | Random Forest | 24.13 | 0.65 | 0.82 | 0.06 |
| +-35%c | XGBoost | 25.25 | 0.61 | 0.83 | 0.07 |
| +-40%c | SVM | 26.00 | 0.66 | 0.81 | 0.05 |
| +-25%c | KNN | 26.63 | 0.62 | 0.85 | 0.08 |
| +-30%c | KNN | 26.63 | 0.64 | 0.83 | 0.07 |
| +-40%c | XGBoost | 26.88 | 0.62 | 0.79 | 0.05 |
| +-45%c | SVM | 28.25 | 0.65 | 0.77 | 0.04 |
| +-5%c | Naive Bayes | 28.57 | 0.64 | 0.79 | 0.05 |
| +-40%c | Random Forest | 28.63 | 0.65 | 0.79 | 0.05 |
| +-35%c | KNN | 32.25 | 0.62 | 0.78 | 0.06 |
| +-50%c | SVM | 32.38 | 0.64 | 0.76 | 0.04 |
| +-45%c | XGBoost | 32.63 | 0.61 | 0.76 | 0.05 |
| +-50%c | XGBoost | 32.63 | 0.59 | 0.78 | 0.06 |
| +-45%c | Random Forest | 33.00 | 0.62 | 0.77 | 0.05 |
| +-10%c | Naive Bayes | 34.75 | 0.57 | 0.81 | 0.09 |
| +-50%c | Random Forest | 36.38 | 0.62 | 0.76 | 0.05 |
| +-40%c | KNN | 37.50 | 0.62 | 0.75 | 0.05 |
| +-45%c | KNN | 39.00 | 0.60 | 0.72 | 0.04 |
| +-15%c | Naive Bayes | 41.13 | 0.57 | 0.75 | 0.07 |
| +-50%c | KNN | 41.88 | 0.59 | 0.71 | 0.04 |
| +-20%c | Naive Bayes | 43.38 | 0.54 | 0.76 | 0.07 |
| +-25%c | Naive Bayes | 44.13 | 0.57 | 0.72 | 0.06 |
| +-30%c | Naive Bayes | 44.25 | 0.57 | 0.71 | 0.05 |

*(Continued)*

**Table 3.** (Continued)

| Scenario | Method | Average AUC Rank | Min AUC Value | Max AUC Value | Standard Deviation AUC Value |
|---|---|---|---|---|---|
| +-35%c | Naive Bayes | 45.50 | 0.57 | 0.68 | 0.04 |
| +-40%c | Naive Bayes | 47.13 | 0.57 | 0.67 | 0.04 |
| +-45%c | Naive Bayes | 47.63 | 0.56 | 0.66 | 0.04 |
| +-50%c | Naive Bayes | 48.75 | 0.55 | 0.64 | 0.04 |

**Table 4. Minimum, mean and maximum AUC value for each combination of drug and algorithm, averaged across all subsampling scenarios.**

| Drug | Method | Min | Mean | Max |
|---|---|---|---|---|
| Gefitinib | SVM | 0.70 | 0.80 | 0.93 |
| Gefitinib | Random Forest | 0.69 | 0.77 | 0.89 |
| Gefitinib | Naive Bayes | 0.58 | 0.63 | 0.74 |
| Gefitinib | KNN | 0.65 | 0.75 | 0.86 |
| Gefitinib | XGBoost | 0.68 | 0.77 | 0.88 |
| Cisplatin | SVM | 0.66 | 0.78 | 0.88 |
| Cisplatin | Random Forest | 0.65 | 0.76 | 0.86 |
| Cisplatin | Naive Bayes | 0.59 | 0.63 | 0.73 |
| Cisplatin | KNN | 0.60 | 0.72 | 0.84 |
| Cisplatin | XGBoost | 0.69 | 0.78 | 0.87 |
| Paclitaxel | SVM | 0.65 | 0.68 | 0.72 |
| Paclitaxel | Random Forest | 0.64 | 0.69 | 0.72 |
| Paclitaxel | Naive Bayes | 0.54 | 0.58 | 0.61 |
| Paclitaxel | KNN | 0.61 | 0.65 | 0.68 |
| Paclitaxel | XGBoost | 0.58 | 0.67 | 0.73 |
| Temozolomide | SVM | 0.74 | 0.84 | 0.95 |
| Temozolomide | Random Forest | 0.73 | 0.82 | 0.90 |
| Temozolomide | Naive Bayes | 0.63 | 0.69 | 0.76 |
| Temozolomide | KNN | 0.68 | 0.79 | 0.92 |
| Temozolomide | XGBoost | 0.74 | 0.83 | 0.93 |
| Etoposide | SVM | 0.66 | 0.75 | 0.88 |
| Etoposide | Random Forest | 0.63 | 0.71 | 0.89 |
| Etoposide | Naive Bayes | 0.56 | 0.61 | 0.71 |
| Etoposide | KNN | 0.59 | 0.68 | 0.84 |
| Etoposide | XGBoost | 0.66 | 0.74 | 0.86 |
| Gemcitabine | SVM | 0.65 | 0.74 | 0.82 |
| Gemcitabine | Random Forest | 0.66 | 0.72 | 0.78 |
| Gemcitabine | Naive Bayes | 0.56 | 0.59 | 0.73 |
| Gemcitabine | KNN | 0.62 | 0.66 | 0.69 |
| Gemcitabine | XGBoost | 0.67 | 0.73 | 0.79 |
| Docetaxel | SVM | 0.76 | 0.87 | 0.97 |
| Docetaxel | Random Forest | 0.76 | 0.86 | 0.97 |
| Docetaxel | Naive Bayes | 0.64 | 0.72 | 0.81 |
| Docetaxel | KNN | 0.71 | 0.81 | 0.91 |
| Docetaxel | XGBoost | 0.76 | 0.87 | 0.96 |
| Doxorubicin | SVM | 0.64 | 0.70 | 0.80 |
| Doxorubicin | Random Forest | 0.62 | 0.68 | 0.78 |
| Doxorubicin | Naive Bayes | 0.56 | 0.58 | 0.64 |
| Doxorubicin | KNN | 0.59 | 0.65 | 0.79 |
| Doxorubicin | XGBoost | 0.59 | 0.65 | 0.71 |

**Table 5. Regression results for all combinations of subsampling scenarios and algorithms for Gefitinib.**

| Scenario | Method | MAE | RMSE | $R^2$ | Spearman |
|---|---|---|---|---|---|
| +-5%r | SVM | 1.28 | 1.54 | **0.50** | **0.63** |
| +-5%r | Random Forest | 1.61 | 1.83 | 0.31 | 0.51 |
| +-5%r | KNN | 1.54 | 1.96 | 0.18 | 0.46 |
| +-5%r | XGBoost | 1.36 | 1.84 | 0.36 | 0.48 |
| +-10%r | SVM | 1.08 | 1.36 | 0.46 | 0.60 |
| +-10%r | Random Forest | 1.26 | 1.53 | 0.34 | 0.53 |
| +-10%r | KNN | 1.27 | 1.65 | 0.21 | 0.47 |
| +-10%r | XGBoost | 1.17 | 1.56 | 0.31 | 0.50 |
| +-15%r | SVM | 1.11 | 1.37 | 0.35 | 0.57 |
| +-15%r | Random Forest | 1.18 | 1.41 | 0.33 | 0.53 |
| +-15%r | KNN | 1.18 | 1.52 | 0.20 | 0.47 |
| +-15%r | XGBoost | 1.16 | 1.48 | 0.25 | 0.50 |
| +-20%r | SVM | 1.04 | 1.27 | 0.35 | 0.59 |
| +-20%r | Random Forest | 1.11 | 1.32 | 0.30 | 0.53 |
| +-20%r | KNN | 1.10 | 1.42 | 0.18 | 0.48 |
| +-20%r | XGBoost | 1.13 | 1.42 | 0.18 | 0.44 |
| +-25%r | SVM | 0.99 | 1.21 | 0.31 | 0.54 |
| +-25%r | Random Forest | 1.04 | 1.24 | 0.28 | 0.52 |
| +-25%r | KNN | 1.02 | 1.32 | 0.18 | 0.47 |
| +-25%r | XGBoost | 1.03 | 1.26 | 0.26 | 0.51 |
| +-30%r | SVM | 0.92 | 1.14 | 0.31 | 0.54 |
| +-30%r | Random Forest | 0.97 | 1.18 | 0.26 | 0.49 |
| +-30%r | KNN | 0.96 | 1.25 | 0.17 | 0.45 |
| +-30%r | XGBoost | 0.97 | 1.20 | 0.23 | 0.47 |
| +-35%r | SVM | 0.88 | 1.10 | 0.25 | 0.52 |
| +-35%r | Random Forest | 0.93 | 1.14 | 0.21 | 0.45 |
| +-35%r | KNN | 0.92 | 1.20 | 0.10 | 0.40 |
| +-35%r | XGBoost | 0.92 | 1.15 | 0.18 | 0.42 |
| +-40%r | SVM | 0.84 | 1.06 | 0.22 | 0.44 |
| +-40%r | Random Forest | 0.86 | 1.06 | 0.21 | 0.43 |
| +-40%r | KNN | 0.88 | 1.14 | 0.10 | 0.36 |
| +-40%r | XGBoost | 0.88 | 1.10 | 0.16 | 0.39 |
| +-45%r | SVM | 0.79 | 1.01 | 0.21 | 0.44 |
| +-45%r | Random Forest | 0.80 | 1.02 | 0.21 | 0.42 |
| +-45%r | KNN | 0.84 | 1.10 | 0.06 | 0.35 |
| +-45%r | XGBoost | 0.81 | 1.04 | 0.18 | 0.40 |
| +-50%r | SVM | **0.73** | **0.95** | 0.23 | 0.45 |
| +-50%r | Random Forest | 0.74 | **0.95** | 0.22 | 0.43 |
| +-50%r | KNN | 0.78 | 1.02 | 0.10 | 0.36 |
| +-50%r | XGBoost | 0.75 | **0.95** | 0.22 | 0.41 |

Bold font indicates the best-performing combination for each metric.

Across all drugs and metrics, the SVM and Random Forests algorithms performed best for every combination of drug and performance metric (Fig 3). Furthermore, predictive performance was highly consistent for all metrics (S9–S15 Figs). When evaluating the mean RMSE ranked values (where the lowest rank was considered best and represented the lowest RMSE value), the RF and SVM algorithms and the +-50%r scenarios performed best (Table 6), and

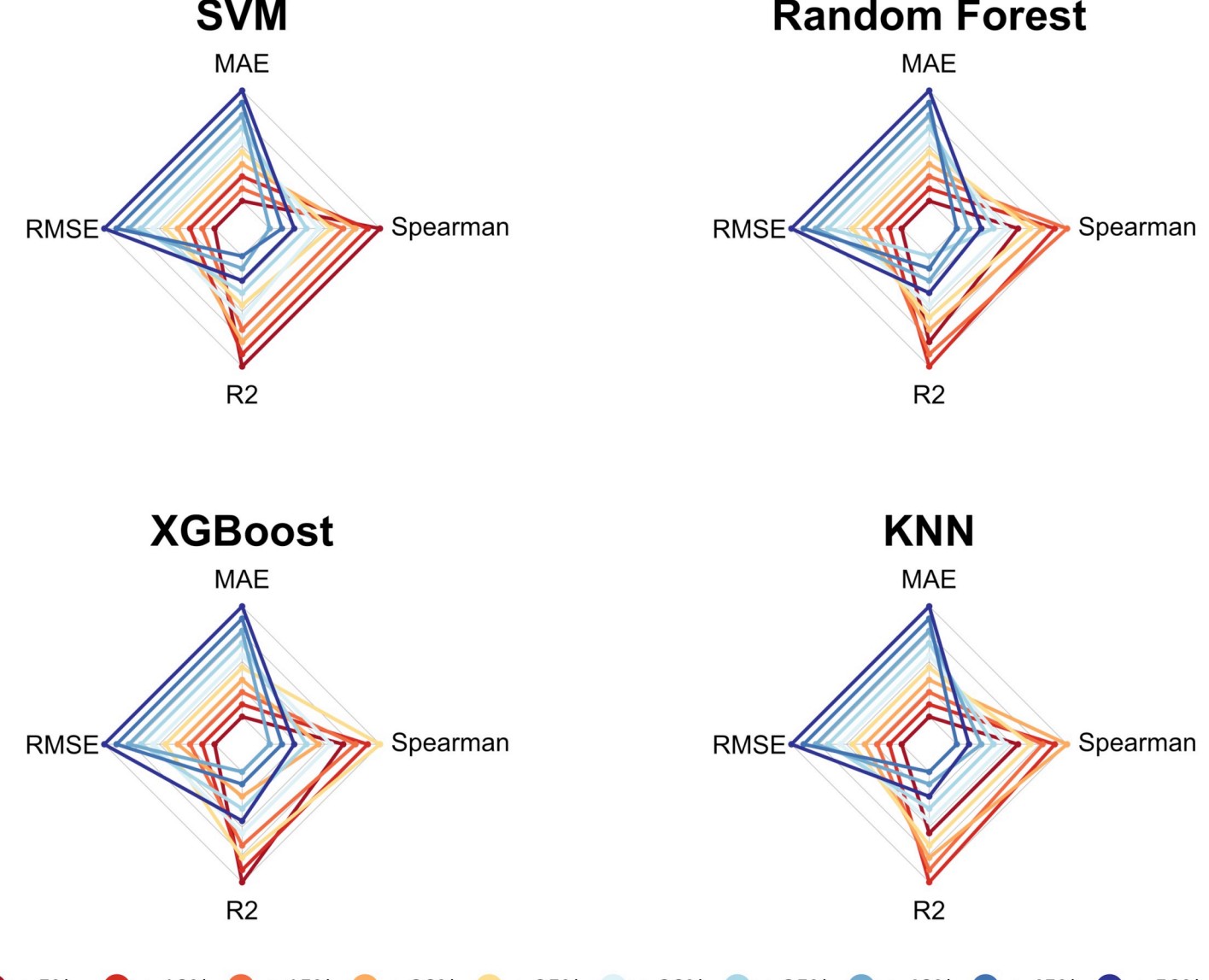

**Fig 3. Gefitinib regression results across four metrics.** These "spider" graphs illustrate how each regression algorithm performed in each subsampling scenario via cross validation on the GDSC cell-line data. Results that are further away from the center represent higher metric values (relatively better performance) than results closer to it. These metrics are RMSE (Root Mean Square Error), MAE (Mean Absolute Error), R-squared and Spearman correlation coefficient. Scenarios that used all cell lines performed best for all algorithms. Specific metric values may be found in Table 5.

predictions for Temozolomide were more accurate overall than those for other drugs (Table 7).

### Classification and regression evaluation

As a way to compare the predictions of the classification versus regression algorithms, we used SCC as a nonparametric measure. For the classification algorithms, we calculated the SCC between the probabilistic predictions that these algorithms produced and the original $IC_{50}$ values. For the regression algorithms we used the SCC values that quantified the correlation between the predicted and actual $IC_{50}$ values. Then for each combination of subsampling

**Table 6. Average RMSE rank for all combinations of subsampling scenarios and algorithms.** RMSE values were ranked for each drug and were then averaged. Lower ranks imply a better result. We also include standard deviation and the minimum and maximum RMSE values.

| Scenario | Method | Average RMSE Rank | Min RMSE Value | Max RMSE Value | Standard Deviation RMSE Value |
|---|---|---|---|---|---|
| +-50%r | Random Forest | **1.50** | **0.67** | **2.53** | **0.61** |
| +-50%r | SVM | 1.75 | 0.68 | 2.56 | **0.61** |
| +-50%r | XGBoost | 2.88 | 0.69 | 2.54 | **0.61** |
| +-45%r | SVM | 4.38 | 0.69 | 2.66 | 0.64 |
| +-45%r | Random Forest | 4.75 | 0.70 | 2.65 | 0.63 |
| +-50%r | KNN | 6.75 | 0.73 | 2.70 | 0.64 |
| +-45%r | XGBoost | 6.88 | 0.73 | 2.67 | 0.64 |
| +-40%r | SVM | 8.00 | 0.72 | 2.77 | 0.67 |
| +-40%r | Random Forest | 8.88 | 0.73 | 2.78 | 0.67 |
| +-45%r | KNN | 10.50 | 0.78 | 2.82 | 0.66 |
| +-40%r | XGBoost | 11.00 | 0.78 | 2.82 | 0.67 |
| +-35%r | SVM | 11.75 | 0.76 | 2.92 | 0.71 |
| +-35%r | Random Forest | 13.00 | 0.76 | 2.94 | 0.71 |
| +-40%r | KNN | 13.88 | 0.81 | 2.94 | 0.69 |
| +-30%r | SVM | 15.38 | 0.80 | 3.07 | 0.75 |
| +-35%r | XGBoost | 15.88 | 0.81 | 3.02 | 0.74 |
| +-30%r | Random Forest | 16.63 | 0.79 | 3.09 | 0.75 |
| +-35%r | KNN | 18.75 | 0.84 | 3.09 | 0.73 |
| +-30%r | XGBoost | 19.25 | 0.84 | 3.17 | 0.77 |
| +-25%r | SVM | 19.88 | 0.80 | 3.25 | 0.81 |
| +-25%r | Random Forest | 21.25 | 0.84 | 3.33 | 0.82 |
| +-30%r | KNN | 21.63 | 0.88 | 3.28 | 0.79 |
| +-20%r | SVM | 23.13 | 0.82 | 3.50 | 0.89 |
| +-25%r | XGBoost | 23.25 | 0.88 | 3.40 | 0.82 |
| +-25%r | KNN | 25.13 | 0.92 | 3.48 | 0.84 |
| +-20%r | Random Forest | 25.75 | 0.90 | 3.55 | 0.89 |
| +-15%r | SVM | 26.88 | 0.86 | 3.57 | 0.91 |
| +-20%r | XGBoost | 28.75 | 0.93 | 3.71 | 0.92 |
| +-20%r | KNN | 29.50 | 0.97 | 3.82 | 0.94 |
| +-15%r | Random Forest | 29.63 | 0.95 | 3.71 | 0.92 |
| +-10%r | SVM | 30.25 | 0.93 | 3.94 | 1.02 |
| +-15%r | KNN | 32.50 | 1.03 | 4.07 | 1.01 |
| +-15%r | XGBoost | 33.13 | 1.06 | 4.00 | 1.02 |
| +-10%r | Random Forest | 33.63 | 1.02 | 4.14 | 1.04 |
| +-10%r | XGBoost | 35.38 | 1.11 | 4.37 | 1.13 |
| +-5%r | SVM | 36.25 | 1.16 | 4.15 | 1.04 |
| +-10%r | KNN | 36.25 | 1.16 | 4.51 | 1.11 |
| +-5%r | Random Forest | 37.50 | 1.28 | 4.30 | 1.01 |
| +-5%r | KNN | 38.88 | 1.35 | 4.47 | 1.01 |
| +-5%r | XGBoost | 39.75 | 1.49 | 4.79 | 1.28 |

Bold font indicates the best-performing combination for each metric.

scenario and drug, we compared the SCC for the same algorithm types against each other (Fig 4). These coefficients were strongly correlated with each other, illustrating that the classification and regression algorithms typically ranked the patients similarly in relation to the original IC$_{50}$ values.

**Table 7. Minimum, mean and maximum RMSE value for each drug and algorithm combination, averaged across all subsampling scenarios.**

| Drug | Method | Min | Mean | Max |
|---|---|---|---|---|
| Gefitinib | SVM | 0.95 | 1.20 | 1.54 |
| Gefitinib | Random Forest | 0.95 | 1.27 | 1.83 |
| Gefitinib | KNN | 1.02 | 1.36 | 1.96 |
| Gefitinib | XGBoost | 0.95 | 1.30 | 1.84 |
| Cisplatin | SVM | 1.04 | 1.36 | 2.14 |
| Cisplatin | Random Forest | 1.04 | 1.38 | 2.11 |
| Cisplatin | KNN | 1.10 | 1.44 | 2.16 |
| Cisplatin | XGBoost | 1.05 | 1.43 | 2.16 |
| Paclitaxel | SVM | 1.87 | 2.50 | 3.56 |
| Paclitaxel | Random Forest | 1.84 | 2.50 | 3.58 |
| Paclitaxel | KNN | 1.95 | 2.64 | 3.74 |
| Paclitaxel | XGBoost | 1.91 | 2.75 | 4.74 |
| Temozolomide | SVM | 0.68 | 0.82 | 1.16 |
| Temozolomide | Random Forest | 0.67 | 0.86 | 1.28 |
| Temozolomide | KNN | 0.73 | 0.95 | 1.35 |
| Temozolomide | XGBoost | 0.69 | 0.93 | 1.49 |
| Etoposide | SVM | 1.80 | 2.30 | 2.93 |
| Etoposide | Random Forest | 1.84 | 2.36 | 2.93 |
| Etoposide | KNN | 1.94 | 2.49 | 3.03 |
| Etoposide | XGBoost | 1.89 | 2.48 | 3.28 |
| Gemcitabine | SVM | 2.56 | 3.24 | 4.15 |
| Gemcitabine | Random Forest | 2.53 | 3.30 | 4.30 |
| Gemcitabine | KNN | 2.70 | 3.52 | 4.51 |
| Gemcitabine | XGBoost | 2.54 | 3.45 | 4.79 |
| Docetaxel | SVM | 1.22 | 1.47 | 1.99 |
| Docetaxel | Random Forest | 1.23 | 1.52 | 2.14 |
| Docetaxel | KNN | 1.34 | 1.69 | 2.74 |
| Docetaxel | XGBoost | 1.25 | 1.55 | 2.23 |
| Doxorubicin | SVM | 1.59 | 2.14 | 3.17 |
| Doxorubicin | Random Forest | 1.58 | 2.16 | 3.28 |
| Doxorubicin | KNN | 1.69 | 2.24 | 3.21 |
| Doxorubicin | XGBoost | 1.61 | 2.25 | 3.51 |

## Informative genes for predicting cell-line responses

The DNA methylation assays target CpG islands associated with genes across the genome. After identifying analysis scenarios that resulted in optimal performance for classification and regression, we used feature ranking to identify genes that were most informative in these scenarios. For the classification analysis, we focused on the +-5%c scenario. For the regression task, we focused on the +-50%r scenario. Table 8 lists the 20 top-ranked genes for Gefitinib. The *CTGF* gene was ranked 1st for the classification analysis and 13th for the regression analysis. The *CTGF* protein plays important roles in signaling pathways that control tissue remodeling via cellular adhesion, extracellular matrix deposition, and myofibroblast activation [78]; these processes are known to influence tumorigenesis and may alter drug responses [79]. For example, EGFR is expressed in many head and neck squamous cell carcinomas and non-small cell lung carcinomas, yet many of these patients do not respond to Gefitinib treatment [80]. This lack of response has been associated with a loss of cell-cell adhesion, elongation of cells,

## Classification vs. Regression Spearman Coefficients

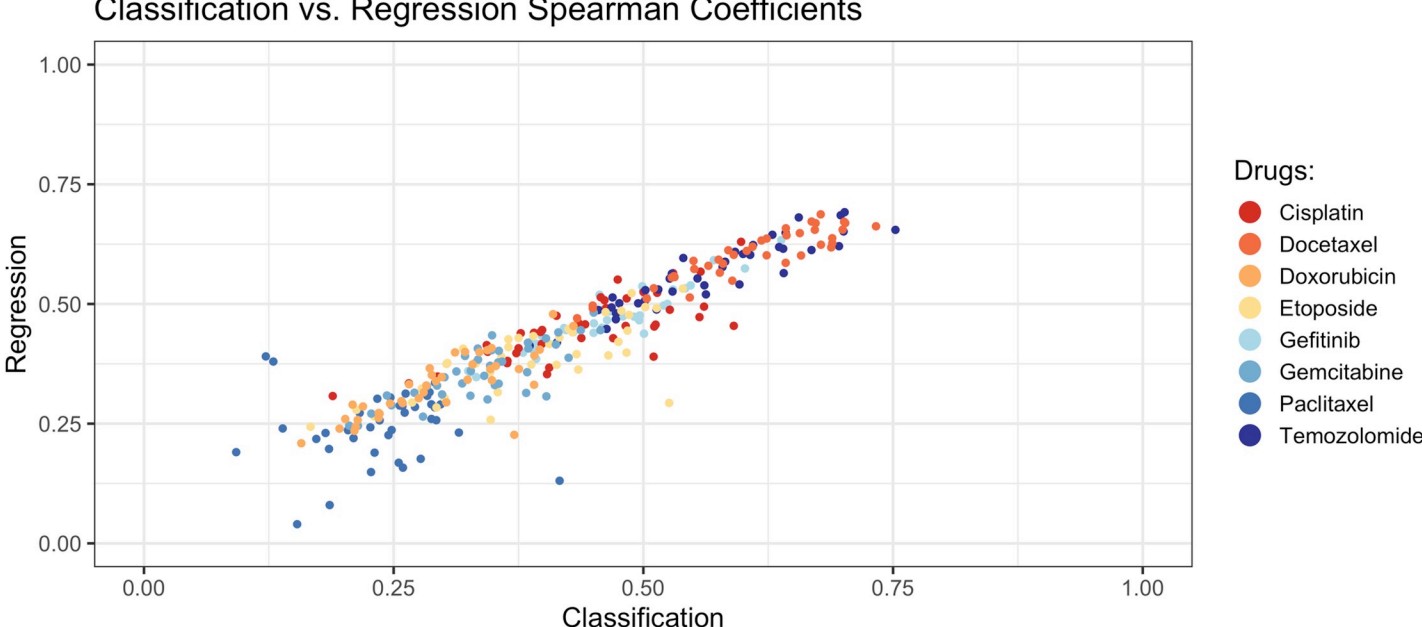

**Fig 4. Spearman correlation coefficient results for classification algorithms (predicted probabilities) and regression algorithms (predicted IC50 values).** For the classification analyses, we calculated the Spearman correlation coefficient between the predicted probabilities and the original $IC_{50}$ values. These are represented on the x-axis. The y-axis represents the Spearman coefficients from the regression analyses. Each dot reflects results for a particular combination of drug, subsampling scenario, and algorithm.

**Table 8. Most informative genes for predicting cell-line responses for Gefitinib.** We used an information-gain analysis to rank genes based on their association with Gefitinib drug response. Genomic coordinates are based on build 37 of the human genome. We used information gain to rank the genes; higher scores indicate more informativeness.

| Classification | | | Regression | | |
|---|---|---|---|---|---|
| *Gene* | *Coordinates* | *Score* | *Gene* | *Coordinates* | *Score* |
| CTGF | chr6:132271356–132271658 | 0.272 | SNAI2 | chr8:49835987–49836231 | 0.060 |
| F11R | chr1:160990718–160991225 | 0.248 | CARD10 | chr22:37914768–37915883 | 0.055 |
| MUM1 | chr19:1354420–1355350 | 0.228 | PTGFRN | chr1:117452203–117453452 | 0.053 |
| RXRB, SLC39A7 | chr6:33167885–33168715 | 0.220 | PNMAL1 | chr19:46974557–46975073 | 0.053 |
| DUSP7 | chr3:52089652–52090845 | 0.204 | A2M, LOC144571 | chr12:9217328–9217715 | 0.052 |
| TFAP2A | chr6:10419399–10420323 | 0.203 | DGKZ | chr11:46366876–46367101 | 0.052 |
| C20orf56 | chr20:22559553–22560001 | 0.201 | SDCBP2 | chr20:1305899–1306554 | 0.052 |
| RAB38 | chr11:87908243–87908614 | 0.201 | ACAP1, KCTD11, TMEM95 | chr17:7254622–7255808 | 0.052 |
| RAB34 | chr17:27044168–27045049 | 0.196 | ANKRD57, SEPT10 | chr2:110370906–110373301 | 0.051 |
| VIM | chr10:17270430–17272617 | 0.192 | SLC44A2 | chr19:10735999–10736396 | 0.050 |
| PAK6 | chr15:40531244–40531589 | 0.192 | ALOX12 | chr17:6898820–6900427 | 0.049 |
| GATA2 | chr3:128215212–128216905 | 0.190 | ZNF625 | chr19:12266998–12267686 | 0.048 |
| SLC9A2 | chr2:103235376–103236554 | 0.188 | CTGF | chr6:132271356–132271658 | 0.048 |
| C20orf56 | chr20:22557517–22559240 | 0.187 | KLF5 | chr13:73632860–73634370 | 0.048 |
| FERMT1 | chr20:6103436–6103970 | 0.186 | NCOR2 | chr12:125003217–125003482 | 0.048 |
| RBM4B | chr11:66444997–66445471 | 0.185 | TBCD, ZNF750 | chr17:80790368–80790581 | 0.047 |
| ORAI2 | chr7:102073605–102074334 | 0.183 | F11R | chr1:160990718–160991225 | 0.046 |
| LOC338799, SETD1B | chr12:122240899–122243390 | 0.181 | OR10H1 | chr19:15918423–15918704 | 0.045 |
| ABHD5 | chr3:43731998–43733108 | 0.181 | PLEK2 | chr14:67878534–67879167 | 0.044 |
| MAZ | chr16:29818681–29819554 | 0.176 | DGUOK | chr2:74153853–74154281 | 0.043 |

and tumor-cell invasion of the extracellular matrix [81–83]. F11R was ranked second in importance for the classification analysis and seventeenth for the regression analysis. The protein encoded by this gene is a junctional adhesion molecule that regulates the integrity of tight junctions and permeability [84]. Although these associations provide some support for our feature-ranking results and that adhesion processes are important to Gefitinib responses, none of the other top-20 genes overlapped between the classification and regression analysis. The lack of agreement between the classification and regression results is not surprising. For example, even though the Random Forests algorithm uses a similar methodology for classification and regression, it is not unlikely that different genes would be selected for classification versus regression. We used data for thousands of genes, and different genes may exhibit similar methylation patterns, so the algorithms may choose different (correlated) genes by random chance. Secondly, the algorithms optimized against different objective functions for classification versus regression; even small differences in how the algorithms prioritized genes could lead to large differences in the gene ranks. However, the SVM and RF models represent multivariate patterns; thus, known cancer genes may alter drug responses in combination with the genes identified via our univariate feature-selection approach, even if they are not among the top-ranked genes.

S15–S21 Tables indicate the top-20 ranked genes for the other 7 drugs. To gain insight regarding the roles that these genes might play in drug responses, we identified gene sets (e.g., pathways, oncogenic signatures) that significantly overlapped with these genes (S22, S23 Tables). For the classification analysis, we identified significant gene sets for 5 drugs (Gefitinib, Cisplatin, Docetaxel, Doxorubicin, Etoposide). Many of these gene sets are associated with cell differentiation, cell-cell communication, and drug resistance; however, these mechanisms did not always align with the respective drugs or target proteins that we expected based on the drugs' known mechanisms. We observed similar patterns for the regression analysis. Two perhaps notable findings are that 1) a gene set associated with EGFR overexpression was associated with Gefitinib responses (this drug targets EGFR) and 2) a gene set associated with Gefitinib resistance was associated with Cisplatin responses, and it has been shown that Cisplatin's ability to induce cell death is dependent in part on EGFR signaling in some cases [85].

### Using methylation profiles from cell lines to predict tumor/patient drug responses

The above analyses used methylation profiles to predict drug responses in cell lines. Via cross validation, we showed that high levels of predictive accuracy are attainable using this approach. We also found that subsampled datasets with more extreme $IC_{50}$ values yielded the best classification results and that the SVM and Random Forests algorithms typically produced the most accurate results. Next we evaluated whether this performance would hold true in a translational-medicine context. The GDSC repository provides methylation profiles for 6,035 tumors from TCGA; these data had been preprocessed using the same methodology as the GDSC samples, thus enabling easier integration and reducing technical biases. For 1,638 TCGA patients, clinical drug-response information was available. These data indicate clinical outcomes over the course of the patients' treatment by physicians (not as part of clinical trials). In many cases, drug-response values for multiple drugs were recorded for a given patient. Each response value was categorized as "clinical progressive disease," "stable disease," "partial response," or "complete response". These respective categories represent increasing levels of response to a given drug.

We trained the SVM and Random Forests classification algorithms on the full GDSC dataset and predicted drug-response categories for each TCGA patient for which methylation and drug-response data were available. Based on our cross-validation results from the GDSC analysis, we focused on the +-5%c and +-10%c scenarios. For each TCGA test sample, our models generated a probabilistic prediction indicating whether that patient would respond to a given drug. We compared these predictions against the ordinal clinical responses for each combination of subsampling scenario (+-5%c and +-10%c), drug, and algorithm (SVM and RF); we calculated the SCC and a corresponding p-value for each comparison and adjusted for multiple tests. Generally, the predictions exhibited low correlation with clinical responses (Table 9); However, the predictions for lower-grade glioma patients who had been treated with Temozolomide were relatively strongly correlated with clinical responses (rho = 0.372; FDR = 0.014), though this result was specific to the Random Forests algorithm and the +-5%c scenario (Fig 5).

**Table 9. Correlation between predicted drug responses based on GDSC cell lines and recorded clinical responses in TCGA patients for selected combinations of subsampling scenarios and algorithms across all drugs.** We treated the clinical drug responses as an ordinal variable and used the Spearman rank correlation coefficient to assess the extent to which the predicted responses correlated with the clinical responses.

| Drug | Scenario | Algorithm | # Samples | Spearman | P-value | FDR |
|---|---|---|---|---|---|---|
| Gefitinib | +-5%c | SVM | 2 | 1.000 | 1.00E+00 | 1.000 |
| Gefitinib | +-5%c | Random Forest | 2 | 1.000 | 1.00E+00 | 1.000 |
| Gefitinib | +-10%c | SVM | 2 | 1.000 | 1.00E+00 | 1.000 |
| Gefitinib | +-10%c | Random Forest | 2 | -1.000 | 1.00E+00 | 1.000 |
| Cisplatin | +-5%c | SVM | 189 | -0.127 | 8.11E-02 | 0.331 |
| Cisplatin | +-5%c | Random Forest | 189 | 0.041 | 5.72E-01 | 0.721 |
| Cisplatin | +-10%c | SVM | 189 | -0.051 | 4.82E-01 | 0.697 |
| Cisplatin | +-10%c | Random Forest | 189 | 0.100 | 1.72E-01 | 0.424 |
| Paclitaxel | +-5%c | SVM | 110 | 0.234 | 1.40E-02 | 0.149 |
| Paclitaxel | +-5%c | Random Forest | 110 | -0.163 | 8.84E-02 | 0.331 |
| Paclitaxel | +-10%c | SVM | 110 | 0.104 | 2.80E-01 | 0.498 |
| Paclitaxel | +-10%c | Random Forest | 110 | -0.073 | 4.48E-01 | 0.697 |
| Temozolomide | +-5%c | SVM | 85 | -0.217 | 4.65E-02 | 0.331 |
| Temozolomide | +-5%c | Random Forest | 85 | 0.372 | 4.53E-04 | 0.014 |
| Temozolomide | +-10%c | SVM | 85 | -0.060 | 5.86E-01 | 0.721 |
| Temozolomide | +-10%c | Random Forest | 85 | 0.176 | 1.07E-01 | 0.343 |
| Etoposide | +-5%c | SVM | 31 | 0.125 | 5.01E-01 | 0.697 |
| Etoposide | +-5%c | Random Forest | 31 | -0.260 | 1.58E-01 | 0.422 |
| Etoposide | +-10%c | SVM | 31 | 0.083 | 6.58E-01 | 0.753 |
| Etoposide | +-10%c | Random Forest | 31 | -0.223 | 2.29E-01 | 0.440 |
| Gemcitabine | +-5%c | SVM | 56 | -0.235 | 8.11E-02 | 0.331 |
| Gemcitabine | +-5%c | Random Forest | 56 | 0.227 | 9.30E-02 | 0.331 |
| Gemcitabine | +-10%c | SVM | 56 | -0.170 | 2.10E-01 | 0.440 |
| Gemcitabine | +-10%c | Random Forest | 56 | 0.207 | 1.25E-01 | 0.364 |
| Docetaxel | +-5%c | SVM | 61 | 0.132 | 3.09E-01 | 0.521 |
| Docetaxel | +-5%c | Random Forest | 61 | -0.158 | 2.25E-01 | 0.440 |
| Docetaxel | +-10%c | SVM | 61 | 0.096 | 4.60E-01 | 0.697 |
| Docetaxel | +-10%c | Random Forest | 61 | -0.155 | 2.34E-01 | 0.440 |
| Doxorubicin | +-5%c | SVM | 61 | -0.237 | 6.56E-02 | 0.331 |
| Doxorubicin | +-5%c | Random Forest | 61 | 0.338 | 7.78E-03 | 0.125 |
| Doxorubicin | +-10%c | SVM | 61 | -0.063 | 6.31E-01 | 0.748 |
| Doxorubicin | +-10%c | Random Forest | 61 | 0.075 | 5.67E-01 | 0.721 |

FDR = Benjamini-Hochberg False Discovery Rate.

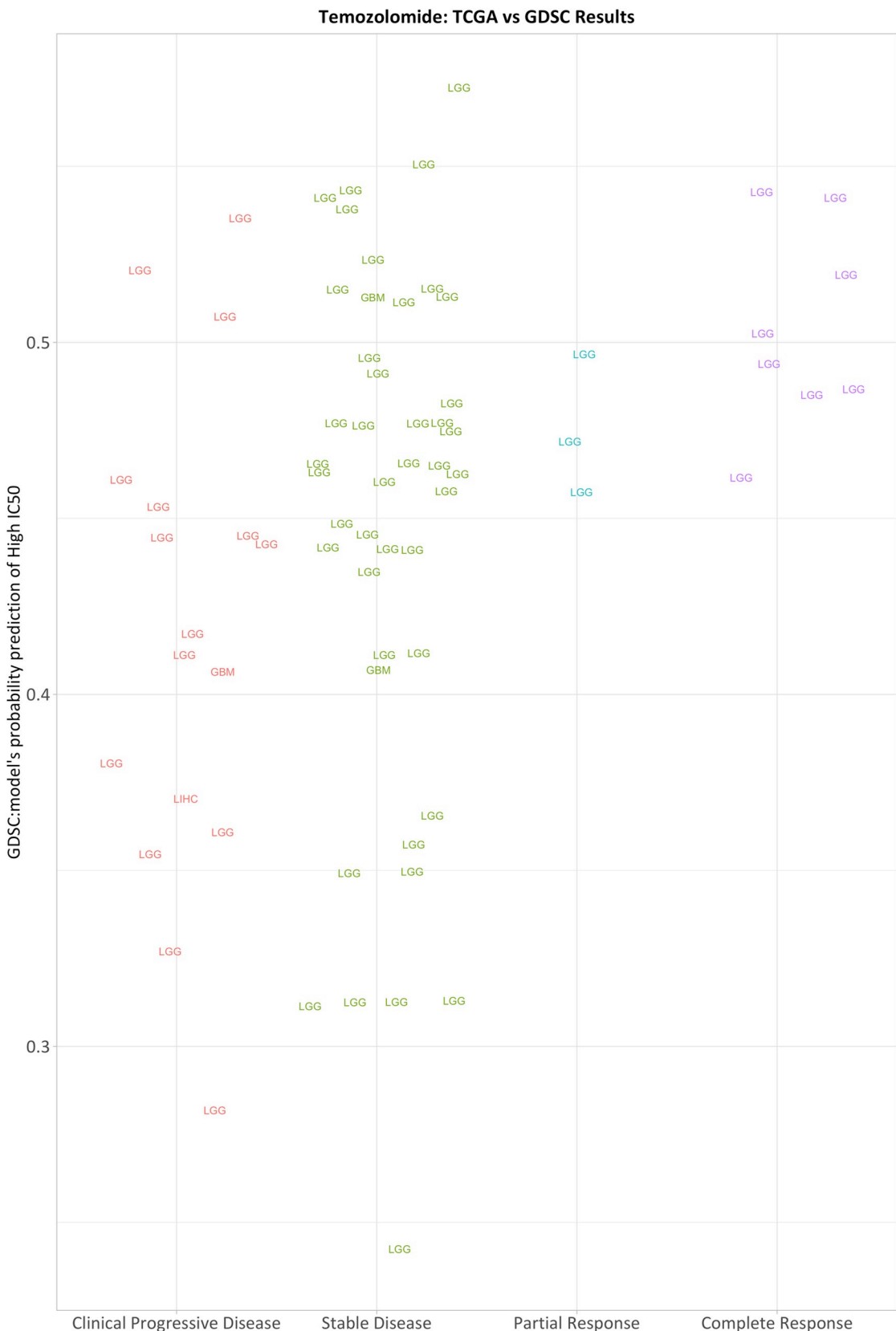

**Temozolomide: TCGA vs GDSC Results**

**Fig 5. Predicting patient drug response from cell-line methylation profiles for Temozolomide (n = 85).** For each TCGA test sample, we used classification models from the GDSC data (+-5%c Random Forest) to generate probabilistic predictions of drug response.

Temozolomide is an oral alkylating agent, is used commonly to treat lower-grade glioma patients, and may reduce seizures and improve prognosis [10].

## Discussion

In an ideal setting, patient data would be used to train predictive models for clinical drug responses directly, as these data may accurately reflect tumor behavior in patients. Environmental factors, the tumor microenvironment, co-existing conditions, and a variety of other factors can affect a tumor's behavior in ways that may not be accounted for in preclinical studies. However, acquiring drug-response data directly from human patients may require conducting many experimental tests on a given patient, which could be unethical, harmful, and subject to many confounding factors. In addition, patients are typically assigned standard-of-care protocols based on their specific cancer type. As a result, experimental drug-response data for large patient cohorts are scarcely available. An alternative approach is to use preclinical samples to identify molecular signatures of drug response and later use those signatures to predict clinical drug responses in patients.

Cell lines serve as preclinical models for drug development. Being able to accurately predict drug responses for a given cell line based on molecular features may help in optimizing drug-development pipelines and explain mechanisms behind treatment responses. We focused on DNA methylation profiles as one type of molecular feature that is known to drive tumorigenesis and modulate treatment responses [47]. When using classification or regression algorithms to predict discrete or continuous responses, respectively, we consistently observed excellent predictive performance when the training and test sets both consisted of cell-line data. Although conventional wisdom advises against discretizing a continuous response variable, where possible, due to loss of information, we wished to evaluate the potential to make effective predictions in this scenario, in part because clinical treatment responses are sometimes represented as discrete values.

Of note, this study focuses primarily on evaluating the effect of subsampling on model performance rather than on introducing new algorithms. Using subsampling, we observed that classification performance generally improved as more extreme examples were used for training and testing, whereas the opposite was often true for the regression analyses. This suggests that during regression, the algorithms benefitted from seeing examples across a diverse range of $IC_{50}$ values for a given drug, whereas the classification algorithms were confounded by seeing examples with relatively similar drug responses, even though sample sizes were smaller. However, again we note that the regression results often differed depending on the evaluation metric used. These results have potential financial implications: if researchers can identify cell lines that are extreme responders for a particular drug, they may only need to generate costly molecular profiles for those cell lines. Future research may elucidate whether this finding generalizes to other types of molecular data and other drugs.

Previous efforts to associate DNA methylation levels with drug responses include work from Shen et al. (2007) [86] who quantified methylation for 32 CpG islands in the NCI-60 cell lines, creating a sensitivity database for ~30k drugs and identifying biomarkers that predict drug sensitivity. Instead, our work uses microarray data to quantify methylation levels for thousands of genes across 987 cell lines but for fewer drugs. Rather than searching for individual genes that predict drug sensitivity, we constructed predictive models that represent

patterns spanning as many as thousands of genes. Such an approach may better represent complex interactions among genes and thus yield improved predictive power, but a tradeoff is reduced model interpretability. We sought to shed some insight into the biological mechanisms that influence drug responses via feature selection, but methods for deriving such insights from genome-wide data are still in their infancy. Recent work using mathematical optimization models shows promise as a way to integrate molecular data from cell lines with drug-sensitivity information to infer resistance mechanisms [87, 88].

A variety of computational methods have been proposed to predict drug responses for cell lines based on molecular data. Classical algorithms like decision trees and support vector machines have been used to predict the clinical efficiency of anti-cancer drugs and to classify drug responses [44, 89–93]. Neural networks [36] and deep neural networks [43] have been used to predict drug response based on genomic profiles from cell lines. Other techniques have included elastic net regression [44, 92, 94], linear ridge regression [45], and LASSO regression [54]. Alternative approaches based on computational linear algebra or network structures have also been applied to infer drug response in cell lines; these include matrix factorization [95], matrix completion [96], and link prediction [97] methods. Finally, a community-based competition assessed the ability to predict therapeutic responses in cell lines using 44 regression-based algorithms [17]. In our study we used diverse algorithms, but our primary focus was data subsampling and evaluating the potential to make accurate predictions of drug response in cell lines using relatively extreme responders, rather than to introduce new algorithms.

We attempted to predict clinical responses for patients from TCGA, but the accuracy of these predictions was typically poor. Integrating datasets can introduce batch effects [98] and other systematic biases; we attempted to mitigate these biases using data that had been preprocessed identically for GDSC and TCGA and using an empirical Bayesian method. However, subtle differences in the way biological samples are handled and processed in the lab can make generalization difficult to achieve. Furthermore, inherent differences between cell lines and tumors may confound such predictions. Cell lines are grown in a controlled environment, and the cells are relatively homogeneous, whereas tumor samples are a heterogeneous milieu of cells. In addition, TCGA tumor responses were based on clinical observations, so there was no direct mapping between these measurements and $IC_{50}$ values for the cell lines. Furthermore, our approach to quantifying predictive performance was different for the GDSC cross-validation analysis compared to the TCGA training/testing analysis. In the former, the class variable represented two possible outcomes (response and non-response). In the latter, the class variable was ordinal. Yet another challenge was that we used cell lines from all available cell types in GDSC. Better accuracy might be attained when training and testing on a single cell type; however, larger sample sizes would be necessary.

Our study has additional limitations that could be addressed in future research. For one, we focused on DNA methylation profiles in isolation, but other types of molecular features likely modulate treatment responses. A number of cell-line studies have used gene-expression profiles to predict drug responses, and future studies could evaluate the potential benefits of incorporating more than one type of molecular feature into response-prediction models. The treatment-response data were often imbalanced, meaning that not all response classes included similar numbers of patients. Hence, additional work could analyze the effect of class imbalance on model performance. Finally, we adjusted the methylation data for dataset and cell type using an empirical Bayesian framework. However, as few as 2–3 samples were available for some of the cell types, so the correction method may have had difficulty adjusting based on such small numbers of examples.

## Conclusion

We applied machine-learning algorithms to predict cytotoxic responses for eight anti-cancer drugs using genome-wide, DNA methylation profiles from 987 cell lines from the Genomics of Drug Sensitivity in Cancer (GDSC) database. We then compared the performance of the classification and regression algorithms and evaluated the effect of sample size on model performance by artificially subsampling the data to varying degrees. The classification algorithms performed best when relatively few cell lines were used to train and test the models, attaining AUC values as high as 0.97. In contrast, the regression algorithms typically performed best when all cell lines were used to train and test the models, though this result depended on the evaluation metric used. For additional validation, we evaluated our ability to train a model based on drug responses in the GDSC cell lines and then accurately predict patient drug responses using data from The Cancer Genome Atlas (TCGA). Because patient-response values are categorical in nature, we only performed classification for these data. In most cases, classification algorithms trained on the full GDSC dataset to predict drug-response categories for TCGA patients were unable to identify patterns in the cell-line methylation data that translated to patient responses.

## Supporting information

**S1 Fig. Example of subsampling process.** When performing classification, we discretized drug-response ($IC_{50}$) values. To evaluate alternative thresholds for discretization, we performed a subsampling analysis. In Scenario 1 illustrated above, we considered the cell lines with the lowest and highest 5% of $IC_{50}$ values. In Scenario 2, we considered the cell lines with the lowest and highest 10% of $IC_{50}$ values. Each scenario used 10% more data than the previous scenario (5% on each side). This pattern continues until all data were considered in the analysis.
(TIF)

**S2 Fig. Graphs for Cisplatin classification analysis.** The graphs compare different scenarios ranked in order of best result. GDSC cell-line data were used to generate ten subsampling scenarios, which we then tested via nested cross validation. Scenarios that are further away from the center represent higher metric values than scenarios closer to it. The evaluated metrics for each algorithm are accuracy (ACC), specificity, recall, Matthews correlation coefficient (MCC), F1 score (F1) and area under the receiver operating characteristic curve (AUC).
(TIF)

**S3 Fig. Graphs for Docetaxel classification analysis.** The graphs compare different scenarios ranked in order of best result. GDSC cell-line data were used to generate ten subsampling scenarios, which we then tested via nested cross validation. Scenarios that are further away from the center represent higher metric values than scenarios closer to it. The evaluated metrics for each algorithm are accuracy (ACC), specificity, recall, Matthews correlation coefficient (MCC), F1 score (F1) and area under the receiver operating characteristic curve (AUC).
(TIF)

**S4 Fig. Graphs for Doxorubicin classification analysis.** The graphs compare different scenarios ranked in order of best result. GDSC cell-line data were used to generate ten subsampling scenarios, which we then tested via nested cross validation. Scenarios that are further away from the center represent higher metric values than scenarios closer to it. The evaluated metrics for each algorithm are accuracy (ACC), specificity, recall, Matthews correlation coefficient (MCC), F1 score (F1) and area under the receiver operating characteristic curve (AUC).
(TIF)

**S5 Fig. Graphs for Etoposide classification analysis.** The graphs compare different scenarios ranked in order of best result. GDSC cell-line data were used to generate ten subsampling scenarios, which we then tested via nested cross validation. Scenarios that are further away from the center represent higher metric values than scenarios closer to it. The evaluated metrics for each algorithm are accuracy (ACC), specificity, recall, Matthews correlation coefficient (MCC), F1 score (F1) and area under the receiver operating characteristic curve (AUC). (TIF)

**S6 Fig. Graphs for Gemcitabine classification analysis.** The graphs compare different scenarios ranked in order of best result. GDSC cell-line data were used to generate ten subsampling scenarios, which we then tested via nested cross validation. Scenarios that are further away from the center represent higher metric values than scenarios closer to it. The evaluated metrics for each algorithm are accuracy (ACC), specificity, recall, Matthews correlation coefficient (MCC), F1 score (F1) and area under the receiver operating characteristic curve (AUC). (TIF)

**S7 Fig. Graphs for Paclitaxel classification analysis.** The graphs compare different scenarios ranked in order of best result. GDSC cell-line data were used to generate ten subsampling scenarios, which we then tested via nested cross validation. Scenarios that are further away from the center represent higher metric values than scenarios closer to it. The evaluated metrics for each algorithm are accuracy (ACC), specificity, recall, Matthews correlation coefficient (MCC), F1 score (F1) and area under the receiver operating characteristic curve (AUC). (TIF)

**S8 Fig. Graphs for Temozolomide classification analysis.** The graphs compare different scenarios ranked in order of best result. GDSC cell-line data were used to generate ten subsampling scenarios, which we then tested via nested cross validation. Scenarios that are further away from the center represent higher metric values than scenarios closer to it. The evaluated metrics for each algorithm are accuracy (ACC), specificity, recall, Matthews correlation coefficient (MCC), F1 score (F1) and area under the receiver operating characteristic curve (AUC). (TIF)

**S9 Fig. Graphs for Cisplatin regression analysis.** We used DNA methylation data from cell lines to predict continuous $IC_{50}$ response values using four regression algorithms. We evaluated the algorithms' performance via nested cross validation for ten subsampling scenarios. Graphs illustrate performance for these scenarios, ranked in order of relative performance for four metrics: RMSE (Root Mean Square Error), MAE (Mean Absolute Error), R-squared and Spearman correlation coefficient. Scenarios further away from the center represent relatively low metric values (and thus better performance). Scenarios that used all cell lines performed best for all algorithms. (TIF)

**S10 Fig. Graphs for Docetaxel regression analysis.** We used DNA methylation data from cell lines to predict continuous $IC_{50}$ response values using four regression algorithms. We evaluated the algorithms' performance via nested cross validation for ten subsampling scenarios. Graphs illustrate performance for these scenarios, ranked in order of relative performance for four metrics: RMSE (Root Mean Square Error), MAE (Mean Absolute Error), R-squared and Spearman correlation coefficient. Scenarios further away from the center represent relatively low metric values (and thus better performance). Scenarios that used all cell lines performed best for all algorithms. (TIF)

**S11 Fig. Graphs for Doxorubicin regression analysis.** We used DNA methylation data from cell lines to predict continuous IC$_{50}$ response values using four regression algorithms. We evaluated the algorithms' performance via nested cross validation for ten subsampling scenarios. Graphs illustrate performance for these scenarios, ranked in order of relative performance for four metrics: RMSE (Root Mean Square Error), MAE (Mean Absolute Error), R-squared and Spearman correlation coefficient. Scenarios further away from the center represent relatively low metric values (and thus better performance). Scenarios that used all cell lines performed best for all algorithms.
(TIF)

**S12 Fig. Graphs for Etoposide regression analysis.** We used DNA methylation data from cell lines to predict continuous IC$_{50}$ response values using four regression algorithms. We evaluated the algorithms' performance via nested cross validation for ten subsampling scenarios. Graphs illustrate performance for these scenarios, ranked in order of relative performance for four metrics: RMSE (Root Mean Square Error), MAE (Mean Absolute Error), R-squared and Spearman correlation coefficient. Scenarios further away from the center represent relatively low metric values (and thus better performance). Scenarios that used all cell lines performed best for all algorithms.
(TIF)

**S13 Fig. Graphs for Gemcitabine regression analysis.** We used DNA methylation data from cell lines to predict continuous IC$_{50}$ response values using four regression algorithms. We evaluated the algorithms' performance via nested cross validation for ten subsampling scenarios. Graphs illustrate performance for these scenarios, ranked in order of relative performance for four metrics: RMSE (Root Mean Square Error), MAE (Mean Absolute Error), R-squared and Spearman correlation coefficient. Scenarios further away from the center represent relatively low metric values (and thus better performance). Scenarios that used all cell lines performed best for all algorithms.
(TIF)

**S14 Fig. Graphs for Paclitaxel regression analysis.** We used DNA methylation data from cell lines to predict continuous IC$_{50}$ response values using four regression algorithms. We evaluated the algorithms' performance via nested cross validation for ten subsampling scenarios. Graphs illustrate performance for these scenarios, ranked in order of relative performance for four metrics: RMSE (Root Mean Square Error), MAE (Mean Absolute Error), R-squared and Spearman correlation coefficient. Scenarios further away from the center represent relatively low metric values (and thus better performance). Scenarios that used all cell lines performed best for all algorithms.
(TIF)

**S15 Fig. Graphs for Temozolomide regression analysis.** We used DNA methylation data from cell lines to predict continuous IC$_{50}$ response values using four regression algorithms. We evaluated the algorithms' performance via nested cross validation for ten subsampling scenarios. Graphs illustrate performance for these scenarios, ranked in order of relative performance for four metrics: RMSE (Root Mean Square Error), MAE (Mean Absolute Error), R-squared and Spearman correlation coefficient. Scenarios further away from the center represent relatively low metric values (and thus better performance). Scenarios that used all cell lines performed best for all algorithms.
(TIF)

**S1 Table. Classification results for all combinations of subsampling scenarios and algorithms for Cisplatin.** Bold font indicates the best-performing combination for each metric.
(DOCX)

**S2 Table. Classification results for all combinations of subsampling scenarios and algorithms for Docetaxel.** Bold font indicates the best-performing combination for each metric.
(DOCX)

**S3 Table. Classification results for all combinations of subsampling scenarios and algorithms for Doxorubicin.** Bold font indicates the best-performing combination for each metric.
(DOCX)

**S4 Table. Classification results for all combinations of subsampling scenarios and algorithms for Etoposide.** Bold font indicates the best-performing combination for each metric.
(DOCX)

**S5 Table. Classification results for all combinations of subsampling scenarios and algorithms for Gemcitabine.** Bold font indicates the best-performing combination for each metric.
(DOCX)

**S6 Table. Classification results for all combinations of subsampling scenarios and algorithms for Paclitaxel.** Bold font indicates the best-performing combination for each metric.
(DOCX)

**S7 Table. Classification results for all combinations of subsampling scenarios and algorithms for Temozolomide.** Bold font indicates the best-performing combination for each metric.
(DOCX)

**S8 Table. Regression results for all combinations of subsampling scenarios and algorithms for Cisplatin.** Bold font indicates the best-performing combination for each metric.
(DOCX)

**S9 Table. Regression results for all combinations of subsampling scenarios and algorithms for Docetaxel.** Bold font indicates the best-performing combination for each metric.
(DOCX)

**S10 Table. Regression results for all combinations of subsampling scenarios and algorithms for Doxorubicin.** Bold font indicates the best-performing combination for each metric.
(DOCX)

**S11 Table. Regression results for all combinations of subsampling scenarios and algorithms for Etoposide.** Bold font indicates the best-performing combination for each metric.
(DOCX)

**S12 Table. Regression results for all combinations of subsampling scenarios and algorithms for Gemcitabine.** Bold font indicates the best-performing combination for each metric.
(DOCX)

**S13 Table. Regression results for all combinations of subsampling scenarios and algorithms for Paclitaxel.** Bold font indicates the best-performing combination for each metric.
(DOCX)

**S14 Table. Regression results for all combinations of subsampling scenarios and algorithms for Temozolomide.** Bold font indicates the best-performing combination for each metric.
(DOCX)

**S15 Table. Informative genes for predicting cell-line responses for Cisplatin.** We used the feature selection to identify informative genes for Cisplatin drug-response prediction. Genomic coordinates are based on build 37 of the human genome. We used information gain to rank the genes; a higher score indicates a more informative gene.
(DOCX)

**S16 Table. Informative genes for predicting cell-line responses for Docetaxel.** We used the feature selection to identify informative genes for Docetaxel drug-response prediction. Genomic coordinates are based on build 37 of the human genome. We used information gain to rank the genes; a higher score indicates a more informative gene.
(DOCX)

**S17 Table. Informative genes for predicting cell-line responses for Doxorubicin.** We used the feature selection to identify informative genes for Doxorubicin drug-response prediction. Genomic coordinates are based on build 37 of the human genome. We used information gain to rank the genes; a higher score indicates a more informative gene.
(DOCX)

**S18 Table. Informative genes for predicting cell-line responses for Etoposide.** We used the feature selection to identify informative genes for Etoposide drug-response prediction. Genomic coordinates are based on build 37 of the human genome. We used information gain to rank the genes; a higher score indicates a more informative gene.
(DOCX)

**S19 Table. Informative genes for predicting cell-line responses for Gemcitabine.** We used the feature selection to identify informative genes for Gemcitabine drug-response prediction. Genomic coordinates are based on build 37 of the human genome. We used information gain to rank the genes; a higher score indicates a more informative gene.
(DOCX)

**S20 Table. Informative genes for predicting cell-line responses for Paclitaxel.** We used the feature selection to identify informative genes for Paclitaxel drug-response prediction. Genomic coordinates are based on build 37 of the human genome. We used information gain to rank the genes; a higher score indicates a more informative gene.
(DOCX)

**S21 Table. Informative genes for predicting cell-line responses for Temozolomide.** We used the feature selection to identify informative genes for Temozolomide drug-response prediction. Genomic coordinates are based on build 37 of the human genome. We used information gain to rank the genes; a higher score indicates a more informative gene.
(DOCX)

**S22 Table. Gene-set analysis for the classification analysis.** We used a statistical overrepresentation test to identify protein classes associated with the top-20 ranked genes in the feature-selection analysis.
(DOCX)

**S23 Table. Gene-set evaluation using GSEA for the regression analysis.** We used a statistical overrepresentation test to identify protein classes associated with the top-20 ranked genes in

the feature-selection analysis.
(DOCX)

## Acknowledgments

The authors express gratitude to patients who donated specimens and data to GDSC and TCGA and those who curated the data and made it publicly available. We used computing resources from the Fulton Supercomputing Laboratory at Brigham Young University to perform these analyses.

## Author Contributions

**Conceptualization:** Sofia P. Miranda, Fernanda A. Baião, Julia L. Fleck, Stephen R. Piccolo.

**Data curation:** Sofia P. Miranda.

**Formal analysis:** Sofia P. Miranda, Julia L. Fleck, Stephen R. Piccolo.

**Methodology:** Sofia P. Miranda, Stephen R. Piccolo.

**Supervision:** Stephen R. Piccolo.

**Writing – original draft:** Sofia P. Miranda, Julia L. Fleck, Stephen R. Piccolo.

**Writing – review & editing:** Sofia P. Miranda, Fernanda A. Baião, Julia L. Fleck, Stephen R. Piccolo.

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
