## [Decision Letter · Decision Letter 0]

29 Sep 2020

PONE-D-20-26313

Predicting drug sensitivity of cancer cells based on DNA methylation levels

PLOS ONE

Dear Dr. Fleck,

Thank you for submitting your manuscript to PLOS ONE. After careful consideration, we feel that it has merit but does not fully meet PLOS ONE’s publication criteria as it currently stands. Therefore, we invite you to submit a revised version of the manuscript that addresses the points raised during the review process.

We look forward to receiving your revised manuscript.

Kind regards,

Tao Huang

Academic Editor

PLOS ONE

Journal Requirements:

Reviewers' comments:

Reviewer's Responses to Questions

**Comments to the Author**

1. Is the manuscript technically sound, and do the data support the conclusions?

Reviewer #1: Yes

Reviewer #2: Yes

Reviewer #3: Partly

Reviewer #4: Yes

Reviewer #5: Yes

2. Has the statistical analysis been performed appropriately and rigorously? 

Reviewer #1: Yes

Reviewer #2: Yes

Reviewer #3: No

Reviewer #4: Yes

Reviewer #5: Yes

3. Have the authors made all data underlying the findings in their manuscript fully available?

Reviewer #1: Yes

Reviewer #2: Yes

Reviewer #3: Yes

Reviewer #4: Yes

Reviewer #5: Yes

4. Is the manuscript presented in an intelligible fashion and written in standard English?

Reviewer #1: Yes

Reviewer #2: Yes

Reviewer #3: Yes

Reviewer #4: Yes

Reviewer #5: Yes

5. Review Comments to the Author

Reviewer #1: This paper applies several machine learning algorithm to the problem of predicting drug sensitivity of cancer cells based on DNA methylation profiles. Overall, the paper is well organized and clearly presented. The following are my main concerns and questions:

(1)This paper does not discuss much on the related work of using machine learning models to predict drug sensitivity. The classification and regression models used in this paper are classic methods. What are the state-of-art methods for predicting drug sensitivity of cancer cells?

(2)In the section regression results, it shows the results for different scenarios, such as +-5%r, +10%r etc. I understand that the scenario in classification (e.g., +-5%c) corresponds to different thresholds to convert continuous outputs to discrete outputs. However, why we need different scenarios in regression task.

(3)In table 8, it shows the informative genes for predicting cell-line response for Gefitinib. It seems that the top-ranked genes for classification task are totally different to regression task. How to explain it? The data and the objective seems the same. The difference is that one formulate it as a classification task and the other formulate it as a regression task.

(4)More detailed discussion may be needed for the conclusion “ classification models derived from cell-line data failed to generalize effectively for tumors”.

Reviewer #2: The authors evaluated machine learning algorithms in predicting drug sensitivity in cell lines using DNA methylation. Although extensive amount of work has been done, the key points and main conclusion of the work are vague, and the impact of this study is doubtful.

1.The main findings should be highlighted in the abstract and the conclusion sections.

2.The lineages in cell lines were overlooked in modeling frameworks and interpretations.

3.The GDSC cell lines were measured on the multi-omic scale, DNA, mRNA and protein. In the introduction section, clearer motivation evaluating only DNA methylation is required.

4.A number of classification algorithms for discretized drug response, and regression approaches for continuous IC50 were evaluated in terms of the predictive accuracy. However evaluation between the two types seems to be missing.

5.Detailed description on feature selection should be included. CTGF gene and etc were highlighted in the paper. Were these appeared using all the algorithms? What was the estimated prediction accuracy based on the algorithms? Did these genes show the high predictive power for all the drugs and all the cell line lineages?

6.For patients’ drug response, the four categories response call was used. When you evaluate the algorithms on the patient data, what was the response call for the cell lines based on IC50? And how do you train the models in the cell lines data? Greater detail is required.

Reviewer #3: -The introduction lacks an appropriate literature review. It is better to review some papers that have used machine learning methods for predicting drug response. Moreover, it is required to present a summary of other features that have been used, such as mutation, copy number variation, and the papers that have used a combination of several types. For example:

1) Moughari, Fatemeh Ahmadi, and Changiz Eslahchi. "ADRML: anticancer drug response prediction using manifold learning." Scientific Reports 10, no. 1 (2020): 1-18.

2) Emdadi, Akram, and Changiz Eslahchi. "DSPLMF: A Method for Cancer Drug Sensitivity Prediction Using a Novel Regularization Approach in Logistic Matrix Factorization." Frontiers in Genetics 11 (2020): 75.

3) Xu, Xiaolu, Hong Gu, Yang Wang, Jia Wang, and Pan Qin. "Autoencoder based feature selection method for classification of anticancer drug response." Frontiers in genetics 10 (2019): 233.

4) Chang, Yoosup, Hyejin Park, Hyun-Jin Yang, Seungju Lee, Kwee-Yum Lee, Tae Soon Kim, Jongsun Jung, and Jae-Min Shin. "Cancer drug response profile scan (CDRscan): a deep learning model that predicts drug effectiveness from cancer genomic signature." Scientific Reports 8, no. 1 (2018): 1-11.

And many more papers.

- ْThe method used for feature selection and identify essential genes is not described well. It is required to give a brief description and intuitive about the method used in this section. Relying on solely naming the used packages is not sufficient, and readers may have no clue of what is going on in this analysis.

- The authors have conducted two biological analyses (1- finding important genes 2- predicting drug response for TCGA tumors). However, both analyzes do not yield significant results. The genes found as important genes for predicting anticancer drug response do not play a significant role in cancer procedures. Moreover, both the regression and classification methods were unable to predict drug response for TCGA tumors. Hence, biological analysis does not confirm model competence. It is required to conduct a more significant biological analysis.

- The authors do not have a significant contribution to the method proposing. They analyze the off-the-shelf machine learning methods in various scenarios.

- Authors have used DNA methylation data instead of gene expression because gene expression data is not stable and is measured using various technologies. It is required to use another type of feature using the same model to determine DNA methylation features' effectiveness.

-The Enrichment of informative genes in GSEA cancer-related gene families can better illustrate the influence of informative genes in cancer-related procedures. Moreover, finding protein complexes related to these genes using CORUM and iRefWeb websites may reveal more information about the drug mechanism.

- The regression analysis to show the correlation of predicted continuous probabilities with ordinal TCGA drug response labels seems inappropriate. The authors can use hypothesis testings that are specifically designed for ordinal variables. Moreover, it is better to apply a pre-processing method to remove batch effects.

- The authors have conducted their analyses on only eight specific drugs. Therefore, the analyses are not comprehensive and may have biased results. It is better to incorporate many more drugs in analyses or select eight randomly drugs.

- The computed criteria for regression methods is not very diverse. RMSE is just the square root of the MSE. Assessing MSE beside RMSE does not convey additional information. Moreover, MAE has almost the same perspective of vision for model evaluation. It is better to compute more diverse criteria such as Pearson correlation coefficient or R^2.

Reviewer #4: The authors conducted an extremely rigorous and interesting study of drug prediction efficacy for a selection of drugs using methylomic data. They tackled the task both as a regression and a classification problem on the popular GDSC dataset and finally tested the translational capabilities of the approach analysing the response prediction on TCGA samples.

Here I include some comments that I think can really help to improve the already good quality of the work:

- Table 1: adding in the supplementary a brief description of the hyperparameters selected for each model could help understanding better the experimental setup. For example the Cost in the SVN is probably referring to the regularization parameter in the cost function, but it is not clear for a reader with limited machine learning knowledge.

- Figures, in general, can benefit from an improved quality, mainly resolution (Figure 2,3). Figure 4 bigger markers and fonts. Also a common style in terms of font type, size and color scheme.

- Regarding the regression analysis it would be very interesting to see reported also the correlation values, to have a better idea on the trend of the predictions beyond the average (MSE, RMSE and MAE).

- This might be a bit beyond the scope of the paper, but as shown, in Costello et al. (https://www.nature.com/articles/nbt.2877.pdf), the expression values (transcriptomic) seem to be more predictive compared to other omic features for the different predictive models considered. I think it would be interesting, if it is not too much work, to repeat the translational analysis with transcriptomic data, to see whether the poor performance observed at methylomic level are consistent or change with a different data type.

Reviewer #5: Title: Predicting drug sensitivity of cancer cells based on DNA methylation levels

Miranda et al. hypothesized that methylation profiles and IC50 drug sensitivity of cancer cell lines can be used to predict cancer patients’ therapeutic outcomes. They extensively trained classification models and regression models using various machine-learning algorithms including SVM and random forest with varying proportions of cancer cell line samples from GDSC cancer cell line panel. It is an interesting observation that a small subset of cancer cell lines with extreme IC50 model yielded the best performance in the development of classification models while did all samples in the development of regression models. However, these models did not predict well patients’ outcomes in a validation study on patients’ data from TCGA. It is well known that there is a nonignorable heterogeneity in molecular profile characteristics between cancer cell lines and patients’ tumors. Observing a poor performance of prediction of patients’ outcome is a bit discouraging, but it also indicates a need of further research to understand and overcome the heterogeneity in cancer cell line-based translational study.

Here are my comments and suggestions.

Page 3

Line 11: Different platform and preprocess algorithms were considered challenging points in combining gene expression datasets. I believe it is also the case in general for combining methylation datasets even though GDSC and TCGA used the same methylation profiling assays.

Page 11

Line 3-6: Performances of various classification algorithms used in this study were well summarized in Table 2 by the proportions of cell lines used for training classifiers of cell lines’ responses to Gemcitabine. However, there is no description nor information about how many methylation loci (or regions) were used in classification and regression analyses.

Page 15

To choose the best combination of classification algorithm and the proportion of cell cell lines, average AUC ranks across drugs were calculated. It would be informative to readers to present the standard deviation and range of observed AUCs along with the AUC ranks in Table 3.

Page 26 and 27.

To examine the association of classification probability with patients’ outcomes, authors used a linear regression. It would be better to display mean and standard deviation of the probability. For an association test, it is suggested to compare means of the probability between four outcome groups.

Table S24: I have a question about Table S24. Is the sample size n in the table the number of cell lines or the number of cancer patients? Please clarify it and I also recommend presenting both numbers in this table.

Page 30

Line 16-18: Authors reported in the result section that using a small (5-10%) of cancer cell lines with extreme IC60s yields the best performance in training a classification model while all samples did in training a regression model. It would be meaningful if you can discuss the reason or insight for this observation because the subsampling of cancer cell lines is the primary focus of this study.

6. PLOS authors have the option to publish the peer review history of their article (what does this mean?). If published, this will include your full peer review and any attached files.

Reviewer #1: No

Reviewer #2: **Yes: **Min Jin Ha

Reviewer #3: **Yes: **Changiz Eslahchi

Reviewer #4: **Yes: **Matteo Manica

Reviewer #5: No

---

## [Author Response · Author response to Decision Letter 0]

1 Dec 2020

Dear editors,

Thank you for taking time to handle this manuscript. We also thank the reviewers for their helpful comments and suggestions, which we have addressed in our revision. We have submitted a separate response letter that contains point-by-point responses to each reviewer's comments. In addition, we have submitted a version of the manuscript that uses the Track Changes feature to show differences between the original version and the revised version.

---

## [Decision Letter · Decision Letter 1]

30 Dec 2020

PONE-D-20-26313R1

Predicting drug sensitivity of cancer cells based on DNA methylation levels

PLOS ONE

Dear Dr. Fleck,

Thank you for submitting your manuscript to PLOS ONE. After careful consideration, we have decided that your manuscript does not meet our criteria for publication and must therefore be rejected.

Specifically:

I am sorry that we cannot be more positive on this occasion, but hope that you appreciate the reasons for this decision.

Yours sincerely,

Tao Huang

Academic Editor

PLOS ONE

Reviewers' comments:

Reviewer's Responses to Questions

**Comments to the Author**

1. If the authors have adequately addressed your comments raised in a previous round of review and you feel that this manuscript is now acceptable for publication, you may indicate that here to bypass the “Comments to the Author” section, enter your conflict of interest statement in the “Confidential to Editor” section, and submit your "Accept" recommendation.

Reviewer #1: All comments have been addressed

Reviewer #3: (No Response)

Reviewer #4: All comments have been addressed

2. Is the manuscript technically sound, and do the data support the conclusions?

Reviewer #1: Yes

Reviewer #3: No

Reviewer #4: Yes

3. Has the statistical analysis been performed appropriately and rigorously? 

Reviewer #1: Yes

Reviewer #3: Yes

Reviewer #4: Yes

4. Have the authors made all data underlying the findings in their manuscript fully available?

Reviewer #1: Yes

Reviewer #3: Yes

Reviewer #4: Yes

5. Is the manuscript presented in an intelligible fashion and written in standard English?

Reviewer #1: Yes

Reviewer #3: Yes

Reviewer #4: Yes

6. Review Comments to the Author

Reviewer #1: My previous concerns has been addressed in this revised version.

Reviewer #3: The authors stated that they aim to evaluated subsampling strategy using traditional machine learning algorithms and they indicate it as their innovation. However, the subsampling strategy is not technically complex enough, nor completely novel. Moreover, it does not yield biologically validated results (see sections two last subsections in result section). The reported SCC for predicting tumor drug responses in TCGA are all below 0.5 (the SCC=1 only occurs for drugs with only 2 samples which can not be considered as a reliable result). The authors mentioned that they report their negative results due to the PLOS policy; nevertheless, it should be noted that there is not any positive biologically verified results. An acceptable paper may contain negative results, when there is adequate persuasive positive results. But the positive results in this paper is restricted to the computational ones and do not have any reliable biological evidence that can verify the meaningfulness of results.

Reviewer #4: The authors addressed all the comments I made in the first revision round, and I think they did a great work in improving their manuscript.

They also motivated very clearly why the analysis of the results using gene expression could be problematic and I totally understand the decision to do not include it in this work.

I noticed also that the authors expanded the coverage of the existing literature by including research works based on deep learning and machine learning. I suggest to take a look also at these works that are quite recent and reports very interesting and accurate results in drug sensitivity prediction: https://doi.org/10.1186/s12859-020-03842-6, https://doi.org/10.1021/acs.molpharmaceut.9b00520.

7. PLOS authors have the option to publish the peer review history of their article (what does this mean?). If published, this will include your full peer review and any attached files.

Reviewer #1: No

Reviewer #3: **Yes: **Changiz Eslahchi

Reviewer #4: **Yes: **Matteo Manica

- - - - -

---

## [Author Response · Author response to Decision Letter 1]

4 Feb 2021

Dear editors:

Thank you for taking time to handle our manuscript entitled, "Predicting drug sensitivity of cancer cells based on DNA methylation levels." We are also thankful to the reviewers for their helpful comments and suggestions. The purpose of this letter is to appeal the recent rejection decision that we received on this manuscript.

After our initial submission, we received helpful comments from 5 different reviewers. We carefully addressed these comments and resubmitted the manuscript on November 30, 2020 (we pasted a copy of these comments at the bottom of this document for your convenience). The academic editor (Dr. Tao Huang) sent our responses to the reviewers and 3 of them replied with additional comments. Reviewer #1 was fully satisfied with the manuscript. Reviewer #3 (Dr. Changiz Eslahchi) re-expressed concerns about 2 particular aspects of the manuscript. It appears that the academic editor rejected the manuscript on the basis of these concerns. Reviewer #4 (Dr. Matteo Manica) suggested that we cite two additional papers. We would be happy to add these two citations.

Below we address Reviewer #3's concerns in a point-by-point response. We do not agree that these concerns are grounds for rejecting the manuscript. Furthermore, we are confident that our manuscript meets all seven of PLOS One's Criteria for Publication.

Comments from Reviewer #3 (Dr. Changiz Eslahchi) and our responses:

"The authors stated that they aim to evaluated subsampling strategy using traditional machine learning algorithms and they indicate it as their innovation. However, the subsampling strategy is not technically complex enough, nor completely novel."

Our response: In his original review, the reviewer commented that "The authors do not have a significant contribution to the method proposing. They analyze the off-the-shelf machine learning methods in various scenarios."

In our reviewer response, we stated that, "The main focus of this research was to study the effect of subsampling on model performance rather than to contribute new algorithms to the field. We evaluated the subsampling strategy using traditional machine learning algorithms as a way to estimate the overall potential to predict drug responses from methylation-based cell-line data. Future studies could explore the potential to improve predictive performance with alternative algorithms or data types."

The reviewer replied that using this subsampling strategy was not "technically complex enough, nor completely novel." We agree that this approach is not technically complex and that others may have applied a similar technique in other contexts. However, technical complexity and technical novelty are not requirements stated in PLOS One's Criteria for Publication. A large proportion of papers in PLOS One use existing methods without necessarily contributing a novel technical innovation. Furthermore, we have applied this subsampling strategy in a research context that is novel and has important implications for the study of biomarker development. As we state in the manuscript, "if researchers can identify cell lines that are extreme responders for a particular drug, they may only need to generate costly molecular profiles for those cell lines."

"Moreover, it does not yield biologically validated results (see sections two last subsections in result section). The reported SCC for predicting tumor drug responses in TCGA are all below 0.5 (the SCC=1 only occurs for drugs with only 2 samples which can not be considered as a reliable result)."

Our response: The reviewer mentioned that only two of our results described in Table 9 resulted in a Spearman Correlation Coefficient (SCC) of 1. This is true, and it is also true that this scenario was from a preliminary analysis that only included external validation data for two tumors. As stated in the manuscript, we included this preliminary result for completeness, but we primarily focused on results for drugs with much larger sample sizes. An SCC value of 1 indicates perfect correlation, and the reviewer implies that we have no "biologically validated results" because none of the other predictions were perfectly correlated with patient responses. However, it is unreasonable to expect perfect correlation. A better indicator of the validity of our predictions is the p-value associated with each correlation test. After multiple-testing correction, we observed one particular result that was statistically significant. When using the Random Forests algorithm to make predictions for the Temozolomide drug (and using the +-5% most extreme responders), our cell-line predictions were correlated with actual patient responses (n = 85) with an SCC of 0.372, a nominal p-value of 4.53e-04, and a False Discovery Rate of 0.014. Admittedly, our prediction model performed poorly on the tumor data in most cases; we acknowledge this in the Discussion section and provide additional context. Furthermore, we did obtain convincing results via cross validation on the cell-line data, even if relatively few results generalized successfully from cell lines to tumors (a challenging task in any cell-line study).

"The authors mentioned that they report their negative results due to the PLOS policy; nevertheless, it should be noted that there is not any positive biologically verified results. An acceptable paper may contain negative results, when there is adequate persuasive positive results. But the positive results in this paper is restricted to the computational ones and do not have any reliable biological evidence that can verify the meaningfulness of results."

Our response: Firstly, even though many of our results were negative, some of our results were positive (as stated above). It is important to report negative results to reduce publication bias and inform other researchers of all tests that were performed. The reviewer questions whether there were enough positive results, but that is subjective. Regardless, the PLOS One Criteria for Publication require methodological soundness but do not mention a requirement that a certain proportion of results be positive. Below we emphasize these points by indicating how we have met each of these criteria.

PLOS One's Criteria for Publication and our responses:

1. The study presents the results of original research. Our manuscript describes original research and places it in context with other studies that have been performed previously. The Introduction and Discussion sections highlight how this study differs from prior studies.

2. Results reported have not been published elsewhere. We have not published these results elsewhere.

3. Experiments, statistics, and other analyses are performed to a high technical standard and are described in sufficient detail. We are confident that we have met this standard. In our responses to the original reviews, we carefully addressed the reviewers' comments. These changes enabled us to improve our analyses considerably. Reviewer #3's comments do not focus on our methodology but rather a perceived lack of novelty. Moreover, in this second round of revisions, all three reviewers answered "Yes" to the question, "Has the statistical analysis been performed appropriately and rigorously?".

4. Conclusions are presented in an appropriate fashion and are supported by the data. We are confident that we have met this standard. Our comments above address Reviewer #3's concern about a perceived lack of novelty.

5. The article is presented in an intelligible fashion and is written in standard English. We are confident that we have met this standard. The reviewers expressed no concerns about the quality of our writing.

6. The research meets all applicable standards for the ethics of experimentation and research integrity. We are confident that we have met this standard. The reviewers expressed no concerns about ethics or research integrity. We used publicly available data for this study.

7. The article adheres to appropriate reporting guidelines and community standards for data availability. We are confident that we have met this standard. The data we used and our analysis code are publicly available.

---

## [Decision Letter · Decision Letter 2]

13 Jul 2021

Predicting drug sensitivity of cancer cells based on DNA methylation levels

PONE-D-20-26313R2

Dear Dr. Fleck,

We’re pleased to inform you that your manuscript has been judged scientifically suitable for publication and will be formally accepted for publication once it meets all outstanding technical requirements.

Kind regards,

Thippa Reddy Gadekallu

Academic Editor

PLOS ONE

Additional Editor Comments (optional):

Reviewers' comments:

Reviewer's Responses to Questions

**Comments to the Author**

1. If the authors have adequately addressed your comments raised in a previous round of review and you feel that this manuscript is now acceptable for publication, you may indicate that here to bypass the “Comments to the Author” section, enter your conflict of interest statement in the “Confidential to Editor” section, and submit your "Accept" recommendation.

Reviewer #6: All comments have been addressed

Reviewer #7: All comments have been addressed

2. Is the manuscript technically sound, and do the data support the conclusions?

Reviewer #6: Yes

Reviewer #7: Yes

3. Has the statistical analysis been performed appropriately and rigorously? 

Reviewer #6: Yes

Reviewer #7: Yes

4. Have the authors made all data underlying the findings in their manuscript fully available?

Reviewer #6: Yes

Reviewer #7: Yes

5. Is the manuscript presented in an intelligible fashion and written in standard English?

Reviewer #6: Yes

Reviewer #7: Yes

6. Review Comments to the Author

Reviewer #6: The authors have addressed all the comments. The paper is well organized and well written. It can be accepted for publication.

Reviewer #7: 1. The study presents the results of original research.

2. Results reported have not been published elsewhere.

3. Experiments, statistics, and other analyses are performed to a high technical standard and are described in sufficient detail.

4. Conclusions are presented in an appropriate fashion and are supported by the data.

5. The article is presented in an intelligible fashion and is written in standard English.

6. The research meets all applicable standards for the ethics of experimentation and research integrity.

7. The article adheres to appropriate reporting guidelines and community standards for data availability.

7. PLOS authors have the option to publish the peer review history of their article (what does this mean?). If published, this will include your full peer review and any attached files.

Reviewer #6: No

Reviewer #7: No

---

## [Editor Report · Acceptance letter]

31 Aug 2021

PONE-D-20-26313R2 

Predicting drug sensitivity of cancer cells based on DNA methylation levels 

Dear Dr. Fleck:

I'm pleased to inform you that your manuscript has been deemed suitable for publication in PLOS ONE. Congratulations! Your manuscript is now with our production department. 

Kind regards, 

on behalf of

Dr. Thippa Reddy Gadekallu 

Academic Editor

PLOS ONE